# Decorate3D: Text-Driven High-Quality Texture Generation for Mesh Decoration in the Wild

**Yanhui Guo**[1]   **Xinxin Zuo**[2]   **Peng Dai**[2]   **Juwei Lu**[2]   **Xiaolin Wu**[1]
**Li Cheng**[3]   **Youliang Yan**[2]   **Songcen Xu**[2]   **Xiaofei Wu**[2]
[1]McMaster University, Canada   [2]Noah's Ark Lab, Canada   [3]University of Alberta, Canada
{guoy143,xwu}@mcmaster.ca, lcheng5@ualberta.ca
{xinxin.zuo1,peng.dai,juwei.lu}@huawei.com
{yanyouliang,xusongcen,wuxiaofei2}@huawei.com

## Abstract

This paper presents Decorate3D, a versatile and user-friendly method for the creation and editing of 3D objects using images. Decorate3D models a real-world object of interest by neural radiance field (NeRF) and decomposes the NeRF representation into an explicit mesh representation, a view-dependent texture, and a diffuse UV texture. Subsequently, users can either manually edit the UV or provide a prompt for the automatic generation of a new 3D-consistent texture. To achieve high-quality 3D texture generation, we propose a structure-aware score distillation sampling method to optimize a neural UV texture based on user-defined text and empower an image diffusion model with 3D-consistent generation capability. Furthermore, we introduce a few-view resampling training method and utilize a super-resolution model to obtain refined high-resolution UV textures (2048×2048) for 3D texturing. Extensive experiments collectively validate the superior performance of Decorate3D in retexturing real-world 3D objects. Project page: https://decorate3d.github.io/Decorate3D/.

## 1   Introduction

The recent development of effective neural 3D reconstruction techniques such as neural radiance field (NeRF) [12] has facilitated the creation of realistic digital replicas for real-world 3D scenes. While significant progress has been made in realistic 3D reconstruction, one remaining challenge is how to allow users to edit or retexture the acquired 3D objects or assets in the digital scene. To address this issue, this paper introduces Decorate 3D, a user-friendly approach to editing 3D objects through either easy-to-use manual editing or text prompt guided texture generation, as showcased in Fig. 1.

Since the implicit representations of the NeRF model are tightly coupled, it is not trivial to achieve the aforementioned operations for the 3D scene's decoration. To facilitate the modifiability of the 3D representations, we bake the NeRF model into a triangle mesh representation with a view-dependent texture in UV space, which is called a *Decomposition* phase in this paper. The decoupled geometry and UV texture representation make the subsequent *Decoration* phase more manageable. In the decoration phase, Decorate3D allows flexible texture editing and controllable generation of UV textures with the instruction of prompts.

The success of the decoration phase demands a solution to the text-to-3D synthesis problem, which has been so far restricted by the lack of paired text and 3D assets. Recent advancements in text-to-image models [23, 19, 15] greatly facilitate text-driven 3D editing or generation [32, 1, 16, 11]. Researchers have proposed approaches using pre-trained text-to-2D diffusion models to optimize NeRF in the zero-shot setting via the Score Distillation Sampling (SDS) [16] strategy. Although

impressive results have been achieved, the naive SDS optimization based on 2D diffusion models lacks 3D awareness, resulting in inharmonious textures that misalign with the geometry.

To address the above issues, during the decoration phase of Decorate3D , we carry out a direct update of the global UV texture map via a structure-aware SDS optimization. We employ a pre-trained depth-guided text-to-image latent diffusion model [29, 23] and incorporate the initial decoupled UV texture from the decomposition phase as an additional form of structural guidance. We empirically find that rendering with the straightforwardly optimized UV texture from SDS tends to be noisy with color distortions. The reason is that the optimized UV texture stands for neural features in effect, which produce rendered neural images that necessitate a neural interpreter. We incorporate the idea of deferred neural rendering [31] and synthesize images of different views by forwarding the rendered neural images to the encoder-decoder of the latent diffusion model. However, the decoder's view-by-view synthesis causes jittering effects. To address this issue, a few-view resampling training strategy is proposed to optimize a global UV texture from sparsely sampled view directions. Finally, to further enhance the visual quality, we use a super-resolution diffusion model to improve the resolution, which is directly applied to the UV space.

Our key contributions can be summarized as follows. First, we propose a system pipeline to allow convenient 3D-consistent decoration of 3D objects captured in the wild; Second, with our structure-aware 3D texture generation, few-view resampling training, and super-resolution enhancement, we are able to synthesize high-quality textures aligned well with the geometry; Finally, we demonstrate the effectiveness of Decorate3D on real-world datasets and have achieved superior performance over state-of-the-art approaches.

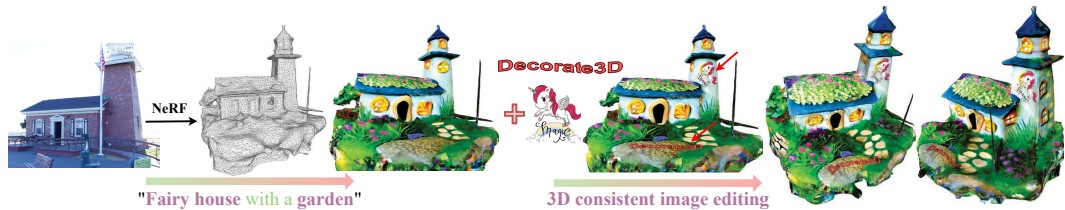

Figure 1: Given captured images of an object, **Decorate3D** supports text-driven high-quality texture generation and user-friendly texture editing. Please zoom in for better visualization.

## 2   Related Work

**Text-to-Image Diffusion Models**    The past two years have witnessed the success of multiple large diffusion models [20, 5, 23, 19, 15] that are able to generate impressive images with photo-realistic details conditioned on an input text prompt. The widely popular stable diffusion [23], was trained on rich paired text-image data and is conditioned on CLIP's [17] frozen text encoder. Beyond text-conditioning, ControlNet [36] extended the stable diffusion by training a parallel hyper-network that allows controllable generation with additional input modalities such as depth maps or edges.

**Text-to-3D Generation Methods**    The development of 2D image generation also greatly facilitates the techniques of text-to-3D generation. CLIP-Mesh [7] proposed a text-driven 3D content generation method using a pre-trained CLIP model. DreamFusion [16] first proposed a score distillation sampling (SDS) method to achieve text-driven optimization for NeRF, relying on a text-to-image diffusion model. Magic3D [9] improved DreamFusion's resolution using a coarse-to-fine optimization strategy. Latent-Paint [11] proposed to bring the NeRF to the latent space and apply the SDS to optimize a latent code for 3D scene generation. More recently, Fantasia3d [3] used a hybrid scene representation of DMTET for SDS optimization, and 3DFuse [27] improved the 3D consistency by incorporating a coarse 3D prior into fine-tuning the diffusion model. Shape-E [6] trained a conditional 3D diffusion model using paired 3D and text data. Concurrent works, TEXTure [22] as well as Text2Tex [2], introduced optimization-free iterative update schemes to paint 3D models from different viewpoints using depth-to-image diffusion models. Although these optimization-free methods may generate plausible results for simple geometries, they usually fail on complex surfaces, resulting in artifacts on the seamed areas. Decorate3D distinguishes itself from previous methods by providing 3D-consistent and high-quality structure-aware texture generation for diverse objects.

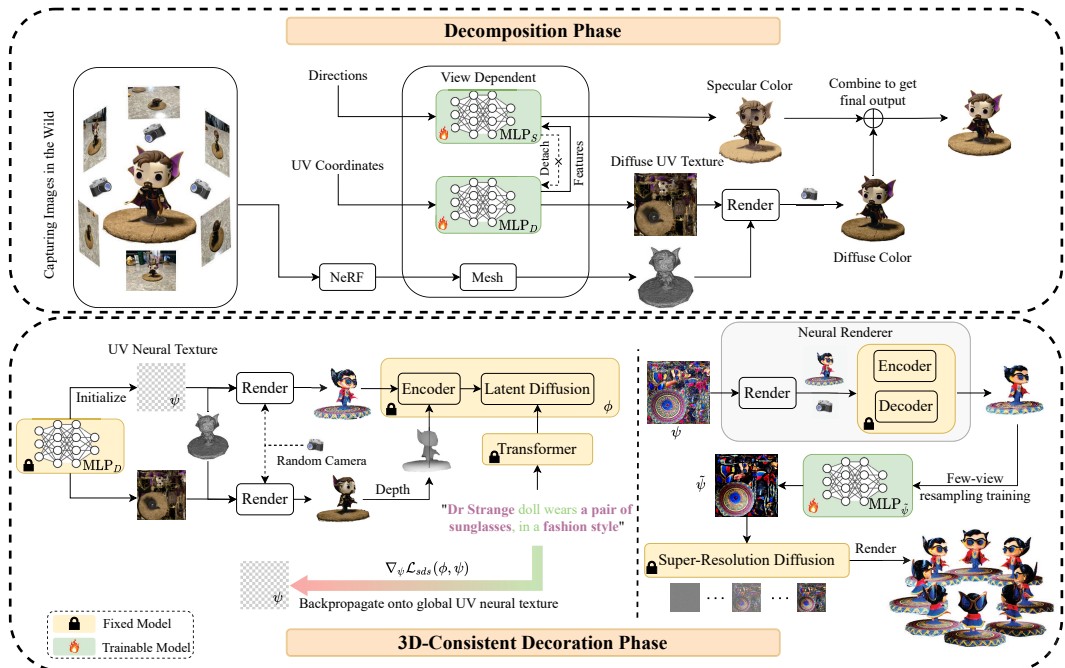

Figure 2: Overview of **Decorate3D** : In the decomposition phase, we utilize NeuS [33] to extract a 3D mesh, and optimize two MLPs (*i.e.* $\mathrm{MLP}_{\mathcal{D}}$ and $\mathrm{MLP}_{\mathcal{S}}$) to model view-dependent textures. In the decoration phase, we initialize the UV neural texture $\psi$ using $\mathrm{MLP}_{\mathcal{D}}$, and optimize the neural texture $\psi$ via the depth-guided diffusion model with the score distillation sampling loss. The neural texture is rendered into sampled views and forwarded to the Neural Renderer to synthesize color images. Afterward, we optimize a MLP through the few-view resampling training method to obtain a UV texture $\tilde{\psi}$ in RGB space. Finally, we enhance the visual quality of $\tilde{\psi}$ with a super-resolution diffusion model.

## 3 Approach

Decorate3D is a two-phase framework to enable 3D-consistent editing of real-world objects. The overall framework of Decorate3D is illustrated in Fig. 2. It consists of a decomposition phase and a decoration phase. Next, we will introduce the decomposition phase in Sec 3.1 followed by the decoration phase. The decoration phase is divided into three parts, including text-driven UV neural texture optimization (Sec 3.2), few-view resampling training (Sec 3.3), and super-resolution on UV texture (Sec 3.4).

### 3.1 Decomposition Phase

Given the captured multi-view images of an object and their respective camera poses obtained by COLMAP [26], we can easily train a NeRF model for novel view synthesis. However, NeRF editing poses challenges due to its tightly coupled representations. To address these challenges and facilitate convenient 3D editing, as well as seamless integration into downstream applications, we propose a solution: decomposing the NeRF representations into a 3D mesh and a view-dependent texture. Specifically, we employ NeuS [33] to reconstruct a triangle mesh $\mathcal{M}$ of the object and calculate the UV atlas associated with the mesh using XAtlas [35]. To represent the real-world texture, we decompose the texture into a diffuse texture modeled by an $\mathrm{MLP}_{\mathcal{D}}$ and a view-dependent specular texture modeled by an $\mathrm{MLP}_{\mathcal{S}}$ conditioned on a given view direction.

Denoting the differentiable mesh rendering by $\mathcal{R}$, we formulate the view rendering process as follows:

$$\mathbf{c}_d, \mathbf{f}_s = \mathrm{sigmoid}(\mathrm{MLP}_{\mathcal{D}}(\mathbf{v})), \tag{1}$$

$$\mathbf{c}_s = \mathrm{MLP}_{\mathcal{S}}(\mathbf{f}_s, \mathbf{d}), \tag{2}$$

$$\mathbf{c} = \mathcal{R}(\mathbf{c}_d + \mathbf{c}_s, \mathcal{M}), \tag{3}$$

where $\mathbf{v}$ refers to 2D UV coordinates after applying positional encoding, and $\mathbf{f}_s$ are the intermediate features to synthesize the specular color. The features $\mathbf{f}_s$ are forwarded to the view-dependent network $\text{MLP}_\mathcal{S}$ to generate view-dependent effects, conditioned on the view direction $\mathbf{d}$. The final rendered image $\mathbf{c}$ is obtained by summing up the diffuse color $\mathbf{c}_d$ and the specular color $\mathbf{c}_s$.

To train the texture networks, we first adopt a typical reconstruction loss $\mathcal{L}_{color}$ to minimize the difference between the rendered image $\hat{\mathbf{c}}(\mathbf{x})$ and its corresponding captured ground truth image $\mathbf{c}(\mathbf{x})$ at each pixel $\mathbf{x}$. The loss function is formulated as follows:

$$\mathcal{L}_{color} = \sum_{\mathbf{x}} \|\hat{\mathbf{c}}(\mathbf{x}) - \mathbf{c}(\mathbf{x})\|^2 \tag{4}$$

In addition, we apply an L1 regularization to enforce sparsity on the specular color:

$$\mathcal{L}_{specular} = \sum_{i} |\mathbf{c_s}(\mathbf{x}_i)| \tag{5}$$

The training of $\text{MLP}_\mathcal{D}$ and $\text{MLP}_\mathcal{S}$ can be achieved with a single-stage framework, as illustrated in the upper part of Fig. 2. More specifically, $\text{MLP}_\mathcal{D}$ and $\text{MLP}_\mathcal{S}$ are jointly optimized, where the gradient from the specular branch will only be used to update $\text{MLP}_\mathcal{S}$, and not be propagated to $\text{MLP}_\mathcal{D}$. The $\text{MLP}_\mathcal{S}$ is a by-product for decomposing the diffuse and specular textures, not used for the decoration phase. We follow instant-ngp [14] to accelerate the optimization to achieve convergence within 3 minutes on a single NVIDIA V100 GPU. More details can be found in the supplementary.

### 3.2 Text-Driven Neural Texture Optimization

The text-driven texture generation is the core element of the decoration phase of Decorate3D . However, diversified 3D generation is often infeasible due to a lack of enough data pairs of text and 3D models. In this context, we utilize the SDS technique for the optimization of the neural texture.

**Preliminary.** Let us first introduce the SDS proposed by DreamFusion [16], which achieves text-to-3D generation based on a pre-trained text-to-image diffusion-based generative model [24]. They utilize a NeRF representation to model the 3D scene. It is a parametric function $\mathbf{x} = \Gamma(\xi)$, which can synthesize an image $\mathbf{x}$ at the desired camera pose. Here, $\Gamma$ is a volumetric renderer, and $\xi$ is an MLP representing a NeRF scene. The diffusion model $\phi$ contains a denoising function $\epsilon_\phi(\mathbf{x}_t; \mathbf{y}, t)$ that predicts the sampled noise given the noisy image $\mathbf{x}_t$ at timestep $t$, and text embedding $\mathbf{y}$. The difference between the noisy image and the denoised image provides the gradient to update $\xi$ such that the density regions with high probability are enforced to match the given text embedding. This gradient update method is named SDS, which is formulated as follows:

$$\nabla_\xi \mathcal{L}_{sds}(\xi) = \mathbb{E}_{t,\epsilon} \left[ w(t)(\epsilon_\phi(\mathbf{x}_t; \mathbf{y}, t) - \epsilon) \frac{\partial \mathbf{x}_t}{\partial \xi} \right], \tag{6}$$

where the noise $\epsilon \sim \mathcal{N}(0, \mathbf{I})$ and $w(t)$ is a weighting function. The neural rendering pipeline $\Gamma(\xi)$ and the diffusion model $\phi$ as modular components of the framework, are amenable to selection. This offers a practical way for text-to-3D synthesis.

**Neural Texture for 3D-Consistent Rendering.** As compared with previous works [16, 11, 9] on optimizing a NeRF model, we directly optimize a neural texture over the UV space. Direct texture optimization is crucial to enforce 3D consistency and preserve texture details. In detail, first we exploit the pre-trained latent diffusion model as the optimization guidance, where it has an encoder $V_e$, a latent diffusion model's denoiser $\epsilon_\phi$, and a decoder $V_d$. In Decorate3D , the neural texture $\psi \in \mathbb{R}^{H \times W \times 3}$ is set to 3 feature channels to fit the requirement of the diffusion model. The gradient calculated via SDS is backpropagated through the diffusion model's encoder to the UV neural texture at a high resolution of $512 \times 512$. Albeit yielding high-resolution images, the optimization computation is reasonable since the optimization exerts an effect on the latent code $\mathbf{z}_t^\psi = V_e(\mathcal{R}(\psi, \mathcal{M}, \mathcal{P}))$ with a resolution of $64 \times 64$, where $V_e$ is the diffusion model's encoder and $\mathcal{P}$ is the sampled camera pose. All network parameters are fixed, while only the neural texture $\psi$ is trainable.

Here, we call the optimized texture *UV Neural Texture* by following the deferred neural rendering [31]. The optimized texture does not act on RGB space, and they are neural features that need a *Neural Renderer* to interpret, which will be introduced shortly.

**Structure-Aware SDS Optimization.** Apart from addressing the 3D-consistency issues, we also want to maintain the coherence between 3D geometry and the generated texture. For example, for a 3D human model, the generated facial texture should be attached to the 3D region corresponding to the face. This structure-aware requirement is overlooked by the naive SDS optimization leading to a multi-face Janus problem. To moderate this issue, Decorate3D utilizes structure-aware SDS optimization by exploiting a depth-guided latent diffusion model [29, 23].

In detail, at each denoising step $t$, we compute the image latent code $\mathbf{z}_t^{\psi}$ produced by the encoder $V_e$ and the depth latent code $\mathbf{z}_t^{\psi_d}$ downsampled from the estimated depth map. They are concatenated as $\mathbf{z}_t$ and forwarded to the latent diffusion module (Eq. 8). Then, following the SDS technique, the gradient of neural texture is computed as follows (Eq. 9):

$$\mathbf{z}_t = [\mathbf{z}_t^{\psi}, \mathbf{z}_t^{\psi_d}], \tag{7}$$

$$\tilde{\epsilon}_\phi(\mathbf{z}_t; \mathbf{y}, t) = \epsilon_\phi(\mathbf{z}_t; t) + \lambda[\epsilon_\phi(\mathbf{z}_t; \mathbf{y}, t) - \epsilon_\phi(\mathbf{z}_t; t)], \tag{8}$$

$$\nabla_\psi \mathcal{L}_{sds}(\phi, \psi) = \mathbb{E}_{t,\epsilon}\left[w(t)(\tilde{\epsilon}_\phi(\mathbf{z}_t; \mathbf{y}, t) - \epsilon)\frac{\partial \mathbf{z}_t}{\partial V_e}\frac{\partial V_e}{\partial \psi}\right], \tag{9}$$

where $\epsilon_\phi$ is the diffusion model's denoiser, $\mathbf{y}$ is the text embedding from the Transformer, and $w(t)$ is a weighting function. Decorate3D uses the classifier-free guidance scheme [15] as stated in Eq. 8, and the guidance weight $\lambda$ for text conditioning is set to 100. To match the depth-guided diffusion model, depth maps are predicted from the rendered views by using the depth estimator [21] of the depth-guided diffusion model rather than the depth buffer of the rendering pipeline. The overall SDS optimization framework is shown at the left bottom of Fig. 2.

In addition to structure-aware SDS optimization, Decorate3D adopts structure-aware initialization. Thanks to the decomposition phase, Decorate3D has an inherent texture that well matches the geometry information extracted from the real-world object. Therefore, Decorate3D can initialize the nerual texture $\psi$ with the output of $\text{MLP}_\mathcal{D}$. This initialization plays a vital role in the optimization process, which can accelerate the convergence as well as provide structure-aware information to help it converge to a structure-aware solution. We verify the effectiveness of the structure-aware techniques in Sec 4.4.

**Decorate3D's Neural Renderer.** As aforementioned, directly optimizing the UV texture with SDS does not yield a traditional RGB UV texture for the rendering pipeline. Fig. 3 shows an example of SDS optimization for text-to-image generation. The optimized texture, *i.e.* neural texture, requires a neural interpreter to convert it back to RGB space after the SDS optimization.

Previous work [11] applies a statistical linear transformation matrix to the optimized UV texture to interpret it in the RGB UV space. However this transformation matrix is suboptimal, and the misalignment will result in color shifts and artifacts.

In Decorate3D , we introduce a neural renderer as the neural interpreter to synthesize photo-realistic images from the optimized neural texture. This rendering process is formulated as follows:

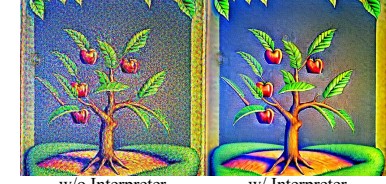

w/o Interpreter          w/ Interpreter

$$I_\mathcal{P} = V(\mathcal{R}(\psi, \mathcal{M}, \mathcal{P})), \tag{10}$$

where $\psi$ is the neural texture, $V$ denotes the network of the neural renderer, and $I_\mathcal{P}$ is the rendered image given the camera pose $\mathcal{P}$. Here, Decorate3D 's neural renderer network $V$ is the encoder-decoder (*i.e.* VAE) module of the depth-guided latent diffusion model [23, 29]. The network $V$ and the traditional renderer $\mathcal{R}$ have amalgamated to form the neural renderer.

Figure 3: The text prompt is "An apple tree". The left shows the image from the direct SDS optimization on a 2D image. The right shows the image after applying interpreter.

## 3.3 Few-View Resampling Training

With the neural texture, Decorate3D can render 3D-consistent high-quality views $I_\mathcal{P}$ through the Neural Renderer with Eq. 10 by sampling different camera poses. But there still exists a practical issue, *i.e.* jittering artifacts, that arises when rendering across different views. This is caused by the view-dependent neural renderer $V$. We empirically find that directly feeding the global UV neural texture $\psi$ to the diffusion model's VAE (*i.e.* $\mathcal{R}(V(\psi), \mathcal{M}, \mathcal{P})$ by swapping the network and renderer

in Eq. 10) can produce the RGB UV texture for the traditional rendering pipeline. But this naive approach usually yields blurry texturing results, as demonstrated in Sec 4.4.

To solve the jittering problem, we devise a few-view resampling (FVR) training method, that transfers the synthesized view-dependent images into a global UV texture $\tilde{\psi}$. As shown in the right bottom of Fig. 2, we use a $\mathrm{MLP}_{\tilde{\psi}}$ to represent the UV texture $\tilde{\psi}$. In the FVR training, we sample $N$ rendered views using the neural renderer with the neural texture $\psi$. The sampled $N$ views should overlay the mesh surface as much as possible. But setting a big $N$ may have negative effects, leading to over-smoothing textures on the overlapped areas that suffer jittering artifacts. The FVR training loss is defined as follows:

$$\mathcal{L}_{FVR}(\tilde{\psi}) = \frac{1}{N} \sum_{i}^{N} \left\| \mathcal{R}(\mathrm{MLP}_{\tilde{\psi}}(\tilde{\mathbf{v}}), \mathcal{M}, \mathcal{P}_i) - V(\mathcal{R}(\psi, \mathcal{M}, \mathcal{P}_i)) \right\|^2, \tag{11}$$

where $\tilde{\mathbf{v}}$ denotes the positional encoding of the 2D UV coordinates of the RGB UV texture $\tilde{\psi}$, and $\mathcal{P}_i$ denotes the $i$-th sampled camera pose. The pose is sampled in spherical coordinates, with two elevation angles $\theta_{cam}$ chosen from $\{-20°, +20°\}$, four azimuth angles $\beta_{cam}$ uniformly sampled between $[0°, 360°]$, and an appropriate viewing distance $r_{cam}$. In our experiments, $N$ is set to 8.

### 3.4 Super-Resolution on UV Texture

From the SDS approach, we can only synthesize images at a resolution of $512 \times 512$. To further improve the spatial resolution of the UV texture $\tilde{\psi}$, Decorate3D applies a super-resolution (SR) diffusion model with a $\times 4$ scale factor [25, 30] on the UV texture to obtain a $2048 \times 2048$ UV texture. The rationale behind this successful application of UV texture upscaling is that the upsampling operation has a spatial locality, concentrating on local textures such as edges. For this reason, SR models are usually trained on cropped image patches [28, 4, 10] instead of the whole image to increase the training efficiency. This spatial locality of SR allows the SR model trained on natural images to be directly applied to the UV texture. Directly applying the SR operation to the UV texture is free from any jittering issues, and 3D consistency is well preserved.

## 4 Experiments

We compare Decorate3D to the state-of-the-art (SOTA) techniques for text-to-3D texture generation and evaluate its performance from both qualitative and quantitative perspectives. The 3D assets and extended videos are presented in the supplementary material.

### 4.1 Implementation Details

To evaluate our approach, we collect real-world datasets from 14 different objects that vary in complexity, including boxes, monitors, shoes, statues, dolls, humans, and lighthouses. Ten objects are captured in the wild using a smartphone, and four objects are selected from some public real-world datasets [34, 8]. The size of each dataset ranges from 70-300 images, and the images are downsampled to a resolution width of 640. We use the Adam optimizer to optimize the $\mathrm{MLP}_{\mathcal{D}}$, $\mathrm{MLP}_{\mathcal{S}}$ and $\mathrm{MLP}_{\tilde{\psi}}$ with a learning rate of $1 \times 10^{-3}$, and the $\psi$ with a learning rate of $1 \times 10^{-2}$. The neural texture optimization in the decoration phase takes about 2 hours for 100K iterations, and the FVR training takes about 5 minutes for 30K iterations, which are measured on 8 NVIDIA V100 GPUs. Please refer to the supplementary material for more details.

### 4.2 Qualitative Evaluation and Comparison

**3D-Consistent Text-Driven Texture Generation.** In Fig. 4, we show the textured mesh and its corresponding real-world image. It can be observed that Decorate3D is able to produce high-quality textured mesh. As the mesh geometry is fixed, we find only texture-related keywords such as nouns or adjectives will affect the generated results, highlighted with a purple color in the prompt. For instance, in the prompt "An astronaut stands up in the milky way", "astronaut" and "milky way" dominate the generation, but the verb "stands up" and the quantifier "an" do not affect the results.

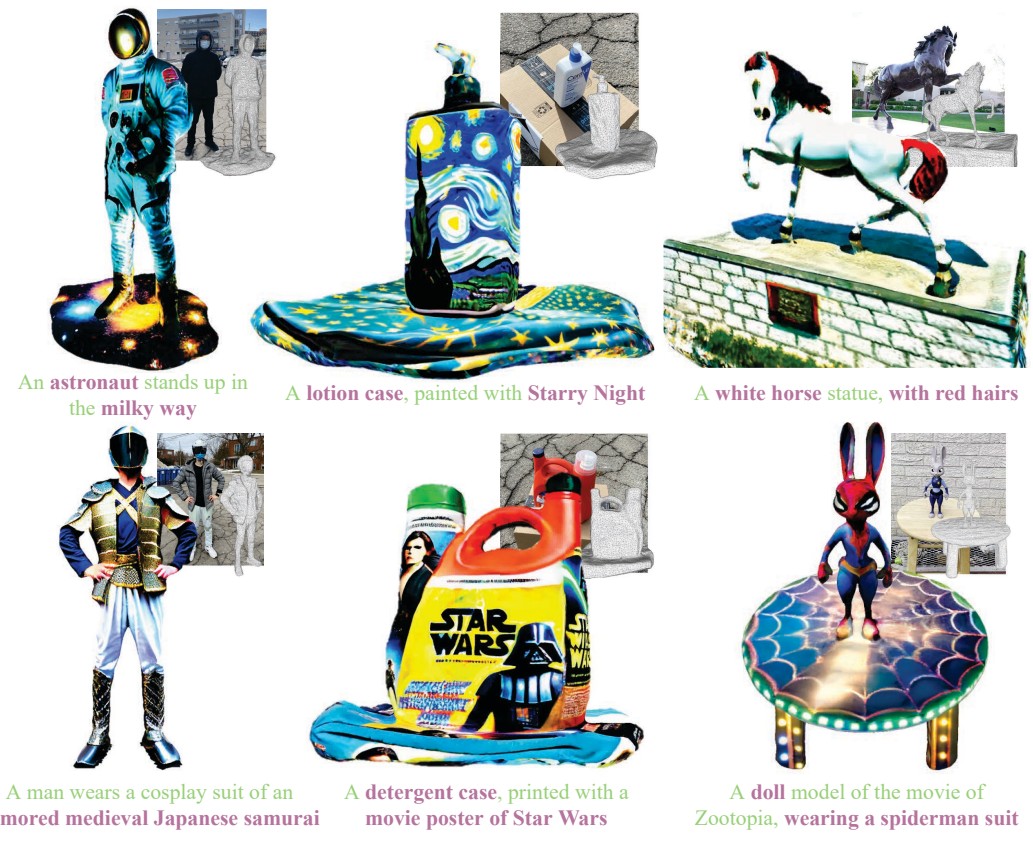

Figure 4: Text-driven texture generation results of Decorate3D.

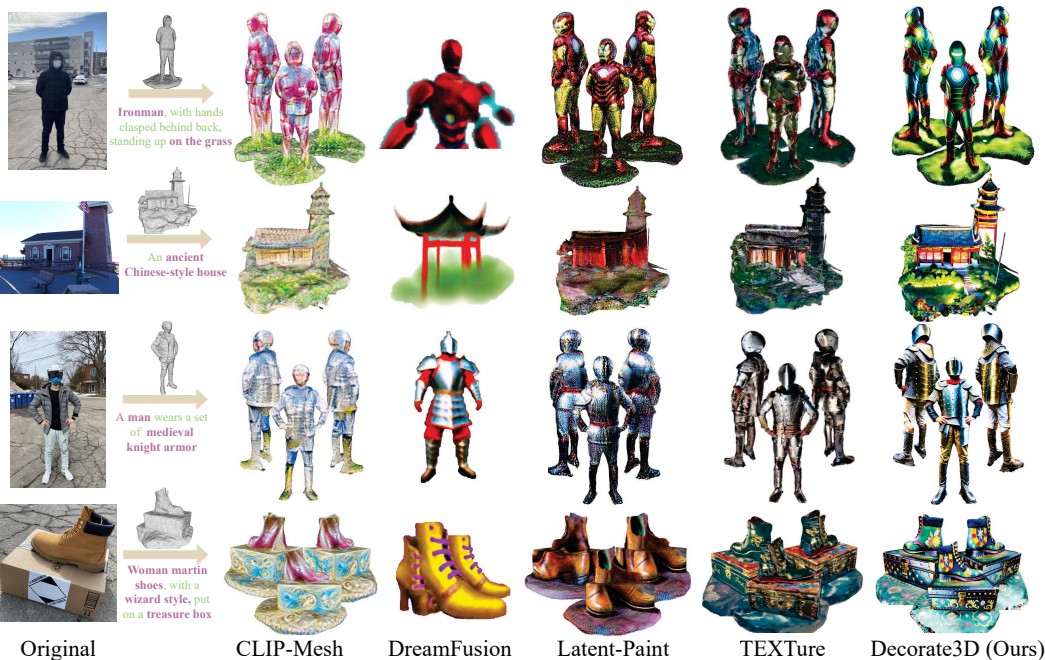

Figure 5: Qualitative comparison with other text-driven texture generation methods, including CLIP-Mesh [7], Latent-Paint [11] and TEXTure [22]. Results from DreamFusion [16] are also shown here. Please zoom in for better visualization.

Fig. 5 shows the qualitative comparison with SOTA text-driven texture generation methods, including CLIP-Mesh [7], Latent-Paint [11] and TEXTure [22]. They all took the reconstructed 3D mesh as input, and the geometry was fixed during texture generation. We can observe that the generated textures from CLIP-Mesh and Latent-Paint lack texture details. The concurrent work TEXTure has some competitive results compared to Decorate3D , such as the results for shoes, because they use a depth-guidance strategy similar to ours. But TEXTure produces weird artifacts on the complex surfaces. For example, in the case of humans, the generated textures from TEXTure have obvious seams, and textures from different parts seem to be glued together and look messy. The artifacts are caused by the iterative update strategy across views. By contrast, the results of Decorate3D have fruitful and clear details. We also present the results from DreamFusion [16]. It tends to produce blurry results that do not actually match the text prompt.

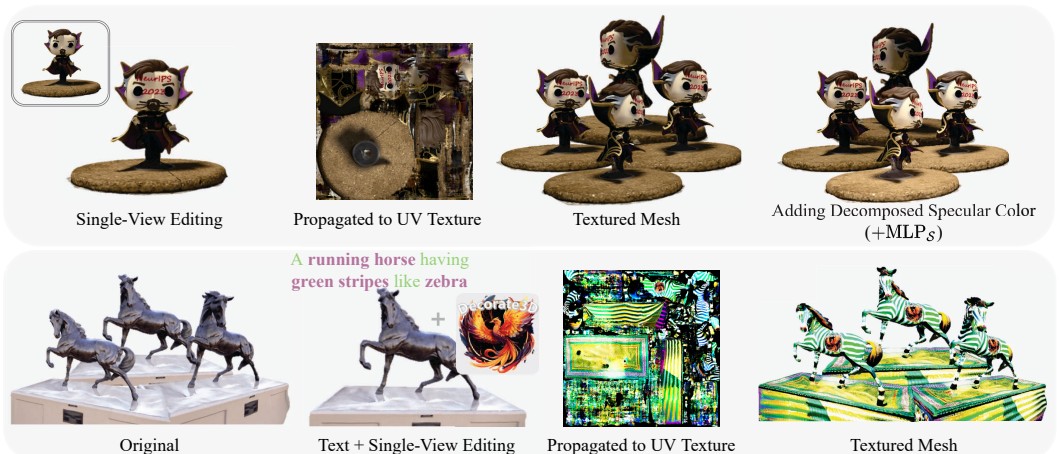

Figure 6: Two examples of applying Decorate3D for 3D-consistent texture editing. As showcased in the first example, a scribble of the "NeurIPS 2023" pattern is painted onto a 2D image (aka. a single rendered view), which is propagated by Decorate3D to the UV texture, updating the 3D-consistent rendered output.

**3D-Consistent Texture Editing.**   Thanks to the decomposed 3D representations, Decorate3D opens up an easy way to edit textures. Fig. 6 depicts a practical and distinctive use case that can be accomplished by Decorate3D . We edit one of the rendered images (not the UV texture), and propagate the editing to the UV texture to synthesize 3D-consistent edited textures from different views. The decomposed view-dependent specular texture can be added back to the textured model. Additionally, Decorate3D allows secondary editing of the text-driven texture generation.

## 4.3   Quantitative Evaluation and Comparison

**CLIP R-Precision Metric.**   Following DreamFusion, we evaluate the CLIP R-Precision [18], an automated metric for the consistency of rendered multi-view images with respect to the input prompt. Here, we calculate the average CLIP score from the front-side, left-side, right-side, and back-side views. We use 70 prompts of 14 objects to generate the test results. For visual quality, we measure the NIQE [13] on rendered images, which is a no-reference image quality assessment. Tab. 1 reports the CLIP R-Precision scores of the compared methods.

**User Studies.**   We invite 44 volunteers to evaluate Decorate3D and its competitors using the mean-opinion-score (MOS) test. For each question, we prepare 5 video results, and the participants are asked to rate the results on a scale from 1 (worst) to 5 (best) based on the overall visual quality of the results and the degree to which they match the text prompt. In the end, we receive 1100 responses from the 44 volunteers. Fig. 7 shows the average MOS scores of the compared methods. As can be seen, Decorate3D is shown to be more favored by human users.

## 4.4   Ablation Studies

**Initialization of UV Neural Texture.**   Fig. 8a shows an ablation study on the initialization of the UV neural texture and Fig. 11 shows more results. As observed, the optimization initialized with

Table 1: Evaluating the correlation of text-driven generated results with their text input using different CLIP models. The CLIP-Mesh's scores may overfit, as it uses the same CLIP model for training and eval. The NIQE is a no-reference image quality evaluation metric.

| Method | R-Precision ↑ | | | NIQE ↓ |
| | CLIP B/32 | CLIP B/16 | CLIP L/14 | |
|---|---|---|---|---|
| CLIP-Mesh [7] | **31.95** $\pm 3.01$ | 30.16 $\pm 2.63$ | 23.99 $\pm 3.71$ | 16.28 $\pm 0.94$ |
| DreamFusion [16] | 29.65 $\pm 4.71$ | 29.63 $\pm 5.11$ | 24.38 $\pm 4.46$ | 16.68 $\pm 3.47$ |
| Latent-Paint [11] | 25.18 $\pm 4.30$ | 26.19 $\pm 2.81$ | 21.22 $\pm 3.03$ | 17.71 $\pm 1.39$ |
| TEXTure [22] | 29.68 $\pm 3.56$ | 28.65 $\pm 2.71$ | 23.15 $\pm 3.31$ | 14.83 $\pm 0.78$ |
| Decorate3D (Ours) | 30.42 $\pm 3.47$ | **30.47** $\pm 3.07$ | **24.89** $\pm 3.41$ | **14.82** $\pm 0.66$ |

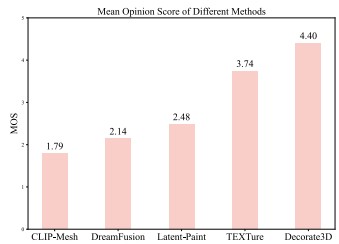

Figure 7: User study results gathered from 44 participants. The results are the average of all responses.

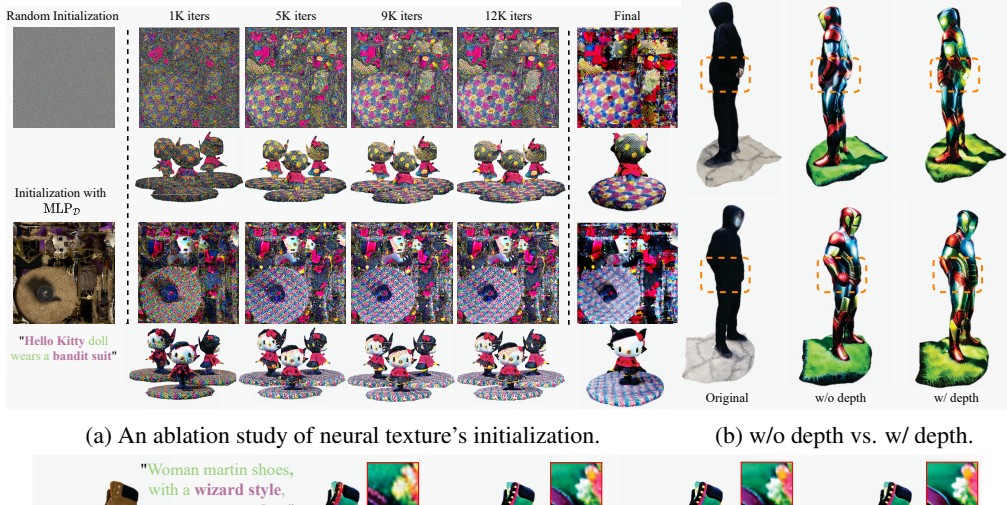

(a) An ablation study of neural texture's initialization.     (b) w/o depth vs. w/ depth.

(c) Ablation studies of FVR training and the super-resolution on UV texture.

Figure 8: Ablation studies of Decorate3D . (a) Random initialization vs. initialization with $\mathrm{MLP}_{\mathcal{D}}$. (b) Naive SDS without depth vs. structure-aware SDS guided by depth. As observed, the naive SDS cannot generate the geometry-matching texture for the man's hands. (c) We compare the rendered results using (1) UV texture yielded by the diffusion model's VAE, (2) Per-view generation, (3) FVR trained $\mathrm{MLP}_{\tilde{\psi}}$, and (4) $\mathrm{MLP}_{\tilde{\psi}}$ +SR.

$\mathrm{MLP}_{\mathcal{D}}$ converges much faster than the random initialization. Moreover, the $\mathrm{MLP}_{\mathcal{D}}$ provides a very strong optimization prior to helping the SDS optimization converge to a better solution that well matches the mesh geometry. For example, initialization using $\mathrm{MLP}_{\mathcal{D}}$ can generate the correct facial texture that fits the geometry, but on the contrary, the random initialization fails.

**Effectiveness of Structure-Aware SDS.** Fig. 8b presents the difference between the naive SDS without depth guidance and the structure-aware SDS with depth guidance, and Fig. 11 shows more results. We can observe that the structure-aware SDS is able to produce geometry-matching textures. For example, the human's arms should be clasped behind the back rather than being akimbo.

**Is the Few-View Resampling Training Necessary?** Fig. 9 compares the per-view generated results of $V(\mathcal{R}(\psi, \mathcal{M}, \mathcal{P}))$ with the results rendered using the FVR-trained UV texture. As observed, results of per-view generation have jittering effects between different views (look at the circle pattern), but results with $\mathrm{MLP}_{\tilde{\psi}}$ have achieved 3D consistency. Fig. 8c (1) presents the results using an alternative solution to eliminate the jittering, *i.e.* directly feeding the neural texture into the diffusion model's VAE by $\mathcal{R}(V(\psi), \mathcal{M}, \mathcal{P})$.

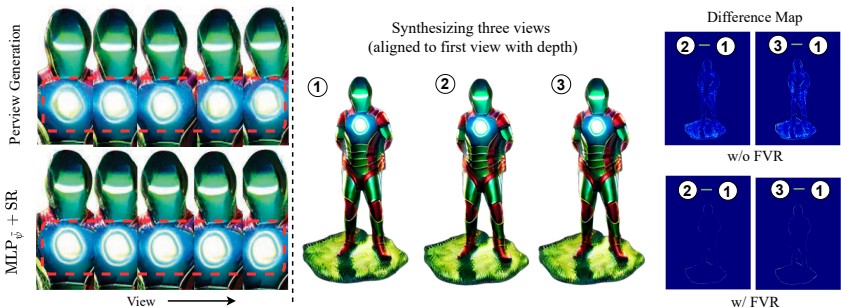

Figure 9: Comparison between the per-view generation (w/o FVR) and rendering with $\mathrm{MLP}_{\tilde{\psi}}$ (w/ FVR). The difference in pixels between neighboring frames is measured after aligning views to the first reference view.

As observed, following this way yields blurry textures. Given an RGB UV texture $\tilde{\psi}$, Decorate3D can directly apply SR on $\tilde{\psi}$. By comparing Fig. 8c (3) and Fig. 8c (4) we can observe more clear results after SR. Fig. 10 shows the effectiveness of FVR training's hyperparameter $N$. Setting a big $N$ for FVR will cause blurry results.

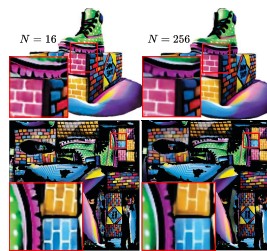

**Effectiveness of Neural Renderer.** As introduced in Sec .3.2, the optimized UV texture by SDS is a neural texture feature. A neural renderer is required to convert the neural UV texture to the RGB UV. The right of Fig. 11 shows the ablation study results on the neural renderer. As observed, the synthesized results without a neural renderer are over-saturated and noisy.

Figure 10: Results with different $N$ for FVR. A bigger $N$ leads to blurry textures.

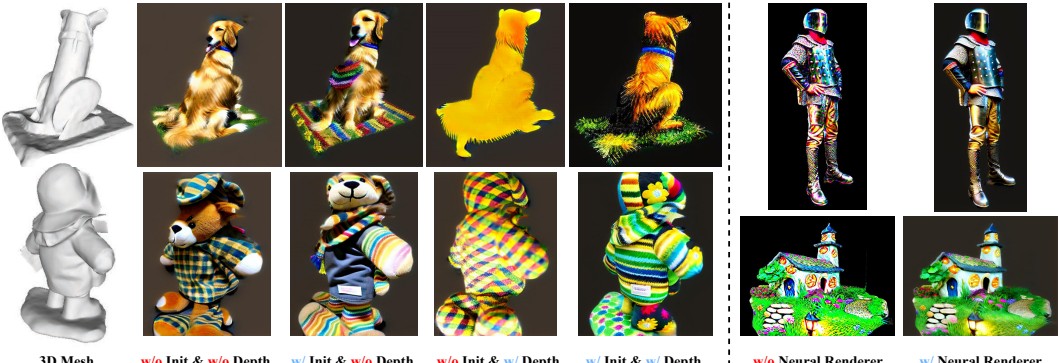

**3D Mesh**    w/o Init & w/o Depth    w/ Init & w/o Depth    w/o Init & w/ Depth    w/ Init & w/ Depth    w/o Neural Renderer    w/ Neural Renderer

Figure 11: Ablation studies on initialization, depth guidance, and neural renderer.

## 5 Conclusion

Decorate3D offers a practical way of decorating real-world 3D models with a user-friendly approach through text-driven texture generation. Our Decorate3D 's techniques, combining directly optimizing UV neural texture, structure-aware optimization, FVR training, and SR enhancement on UV texture, collectively advance the line of 3D texture generation in pursuit of the best possible quality. Extensive experiments demonstrate the superiority of Decorate3D over SOTA methods.

**Limitations** First, the style and quality of the generated texture heavily depend on the pre-trained stable diffusion models. Second, even though Decorate3D adopts structure-aware optimization techniques, the multi-face Janus problem still remains in a flat geometry. For example, Decorate3D cannot distinguish the front and back sides of a monitor object, as shown in our supplementary material. Lastly, Decorate3D does not support jointly optimizing both the geometry and texture, which may result in inconsistency with the mesh geometry. We remain these problems for future study.

## Acknowledgments and Disclosure of Funding

We gratefully acknowledge the support of MindSpore (https://www.mindspore.cn/), CANN (Compute Architecture for Neural Networks) and Ascend AI Processor used for this research.

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
