# Supplementary Material
# Decorate3D: Text-Driven High-Quality Texture Generation for Mesh Decoration in the Wild

**Yanhui Guo**[1]  **Xinxin Zuo**[2]  **Peng Dai**[2]  **Juwei Lu**[2]  **Xiaolin Wu**[1]
**Li Cheng**[3]  **Youliang Yan**[2]  **Songcen Xu**[2]  **Xiaofei Wu**[2]
[1]McMaster University, Canada   [2]Noah's Ark Lab, Canada   [3]University of Alberta, Canada
{guoy143,xwu}@mcmaster.ca, lcheng5@ualberta.ca
{xinxin.zuo1,peng.dai,juwei.lu}@huawei.com
{yanyouliang,xusongcen,wuxiaofei2}@huawei.com

## 1 Project Website of Decorate3D

Please check out the project website https://decorate3d.github.io/Decorate3D/ for more demonstration results.

## 2 More Implementation Details of Decorate3D

### 2.1 MLP Architectures

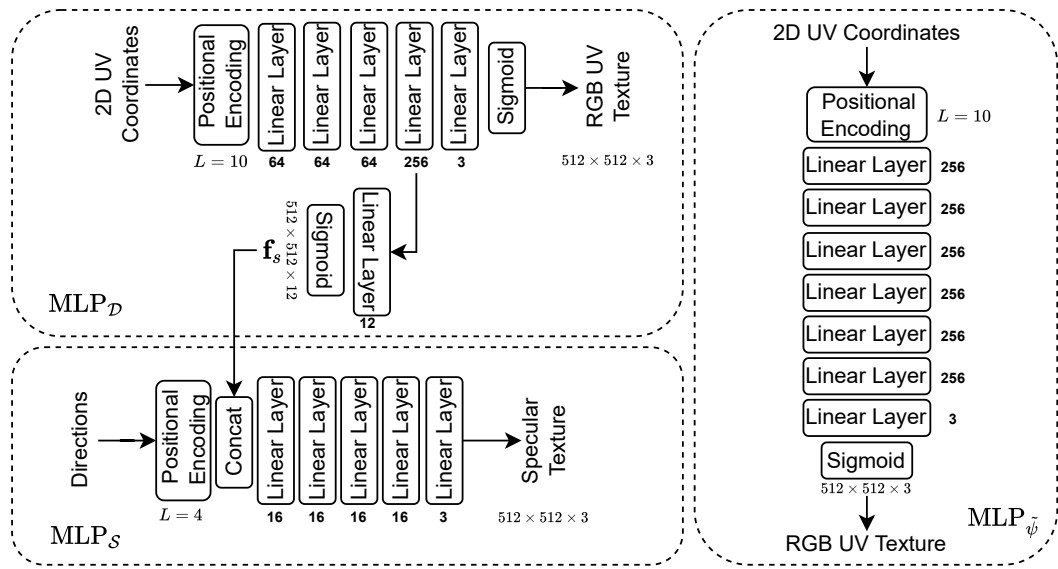

Figure 1: The network structures of $\mathrm{MLP}_{\mathcal{D}}$, $\mathrm{MLP}_{\mathcal{S}}$ and $\mathrm{MLP}_{\tilde{\psi}}$.

Fig. 1 illustrates the MLP network structures of Decorate3D . For the MLP-based neural representations, we adopt the sinusoidal positional encoding function [4] using frequencies $2^0, 2^1, \ldots, 2^{L-1}$, where we set $L = 10$ for both $\mathrm{MLP}_{\mathcal{D}}$ and $\mathrm{MLP}_{\tilde{\psi}}$, and $L = 4$ for $\mathrm{MLP}_{\mathcal{S}}$. For $\mathrm{MLP}_{\mathcal{D}}$, the number of hidden layers is set to 3, and the number of hidden units is set to 64. The number of hidden units of the penultimate layer is set to 256. For $\mathrm{MLP}_{\mathcal{S}}$, the number of hidden layers is set to 4, and the

37th Conference on Neural Information Processing Systems (NeurIPS 2023).

number of hidden units is set to 16. In the hidden layers, we adopt the ReLU activation function. The shared intermediate feature $\mathbf{f}_s$ is set to 12 channels followed by a sigmoid activation function. For $\text{MLP}_{\tilde{\psi}}$, the number of hidden layers is set to 6, and the number of hidden units is set to 256. We use weight normalization [9] to stabilize the training process.

## 2.2 Optimization Settings

The learning rate in the decomposition phase is set to $1 \times 10^{-3}$ and decays by $\gamma = 0.1$ every 10K iterations. We optimize the decomposition phase for 70K iterations with a batch size of 1. For the optimization of neural texture in the decoration phase, we set the batch size to 1 and use the Adam optimizer with a learning rate of $1 \times 10^{-2}$. The learning rate decreases following the values of the cosine function to 0 after 100K iterations. For the FVR training, the learning rate is set to $1 \times 10^{-3}$ and the training involves 30K iterations.

## 2.3 Random Camera Sampling

We follow Poole *et al*. [6] to add random augmentations to the camera during the decoration phase training. At each iteration, a camera pose is randomly sampled in spherical coordinates, with the elevation angle $\theta_{cam} \in [-30°, 45°]$, azimuth angle $\beta_{cam} \in [0°, 360°]$ and the distance from the origin $r_{cam} \in [0.8, 3]$ varying according to the objects. We append view-dependent text to the given input text using a weighted combination of the text embeddings for appending "front view," "left-side view," "right-side view", and "back view" based on the value of the azimuth angle $\beta_{cam}$.

## 2.4 Score Distillation Sampling

In Decorate3D , we adopt the depth2image latent diffusion model from Stable Diffusion v2 [10] to achieve the structure-aware SDS optimization. The encoder-decoder module of the diffusion model is used as the network for the deferred neural renderer. We sample the timestep $t \sim \mathcal{U}(0.02, 0.5)$ for the SDS optimization and set $w(t) = \sigma^2(t)$, where $\sigma^2(t)$ is the noise level at the timestep $t$.

## 2.5 Training Efficiency

In the decomposition phase, to speed up the training, first, we eliminate the computation cost spent on differentiable rendering and pre-compute the mapping from image pixels to UV coordinates. This mapping relationship only needs to be computed once before the training starts. Second, we adopt the acceleration strategies used in instant-ngp [5]. Basically, the two MLPs ($\text{MLP}_{\mathcal{D}}$ and $\text{MLP}_{\mathcal{S}}$) are implemented with tiny-cudann. After the acceleration, the training for the decomposition of the diffuse texture and the view-dependent texture is finished within around 3 minutes on a single NVIDIA V100 GPU.

The neural texture optimization in the decoration phase takes about 2 hours and the FVR training takes about 5 minutes, both measured on 8 NVIDIA V100 GPUs with a batch size of 1 per GPU.

## 2.6 Inference Speed

For the inference, Decorate3D only requires painting the mesh with the color sampled from the RGB UV texture as fast as the traditional rendering pipeline. The rendering speed depends on the rendering software, the total number of polygons, and the texture resolution. For example, given a $2048 \times 2048$ UV texture after super-resolution, Decorate3D takes around 0.73s to render a single image view ($2560 \times 3412$) using Pytorch3D's rendering library [7], which is much faster than synthesizing novel views with the same resolution as that of ours using NeRF representations.

## 3 Real-Wolrd Datasets

We have collected real-world datasets from 14 different objects. Five objects are selected from the public real-world datasets [11, 2]. And the rest are captured using a smartphone. The size of each dataset ranges from 70-300 images, which have been downsampled to a resolution width of 640. Fig. 2 shows two examples of each dataset.

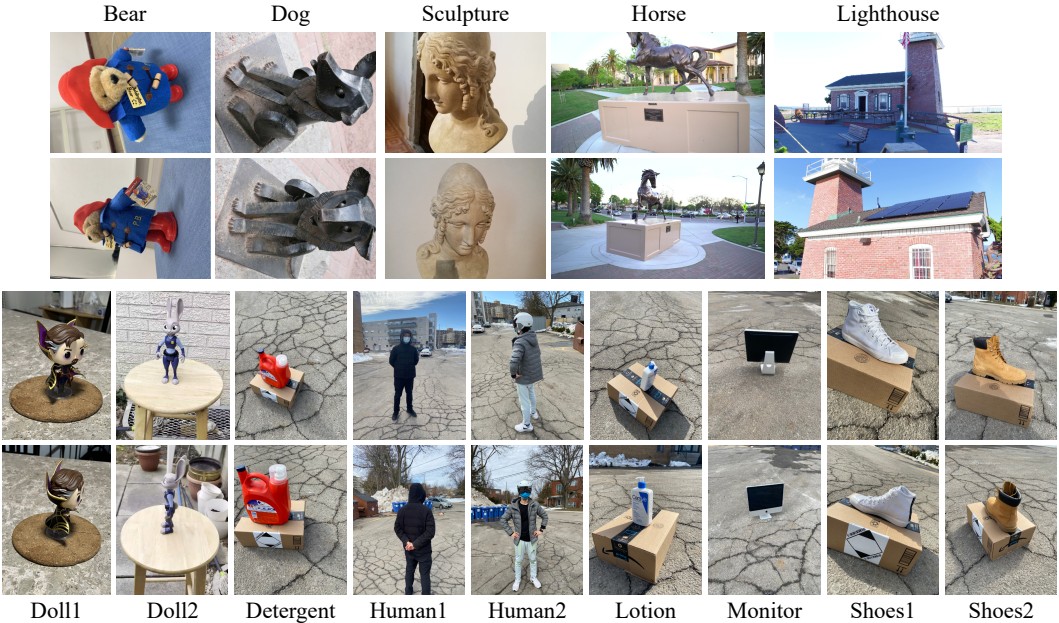

Figure 2: Examples of the collected real-world datasets.

# 4 Details of User Study

Fig. 3 illustrates the questionnaire set up for our user study experiment. We request the volunteers evaluate each video according to its visual quality and degree of alignment with the provided text prompt. The videos are presented in a random order to the human audience for evaluation, and the method names are hidden from the audience.

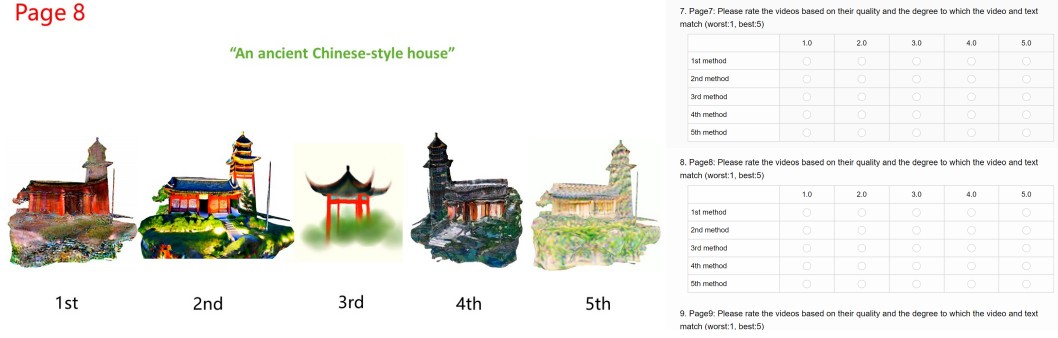

Figure 3: Illustration of the questionnaire of our user study. The left part shows the video results of different methods for visual comparison. The right part is the evaluation form to collect the opinion scores.

# 5 More Qualitative Results

Fig. 4∼Fig. 17 show additional qualitative results for each object in our real-world dataset. As observed, our model is capable of producing high-quality results for both simple and complex geometries of real-world objects.

# 6 More Qualitative Comparison Results

Fig. 18∼Fig. 20 show more qualitative comparison with the state-of-the-art methods including CLIP-Mesh [1], DreamFusion [6], Latent-Paint [3], and TEXTure [8].

# 7 Discussion on Failure Cases

In this section, we show the failure cases of Decorate3D . As mentioned in the main paper, even though Decorate3D adopts the novel structure-aware optimization techniques, the multi-face Janus problem still remains in the flat surfaces. Fig 21 presents a failure case on the "monitor" object. Besides, if the text prompt does match the geometry, the results may deviate from the meanings of the prompts. Fig 22 presents two geometrically misaligned results for the objects "human2" and "shoes1", respectively. As seen, we provide a prompt of "A tiger" such that we hope to obtain a textured mesh of a tiger from both the geometry and texturing perspectives. Due to the fixation of the mesh, the prompt applied to the "human2" object yields misaligned results. Albeit the mismatched geometry, we can also observe that Decorate3D 's paintings are keyword-oriented, allowing the generated textures to reflect the keywords of the given prompt. For example, the textured "human2" object is painted with the tiger's patterns and the "shoes1" object is attached with the cat's patterns.

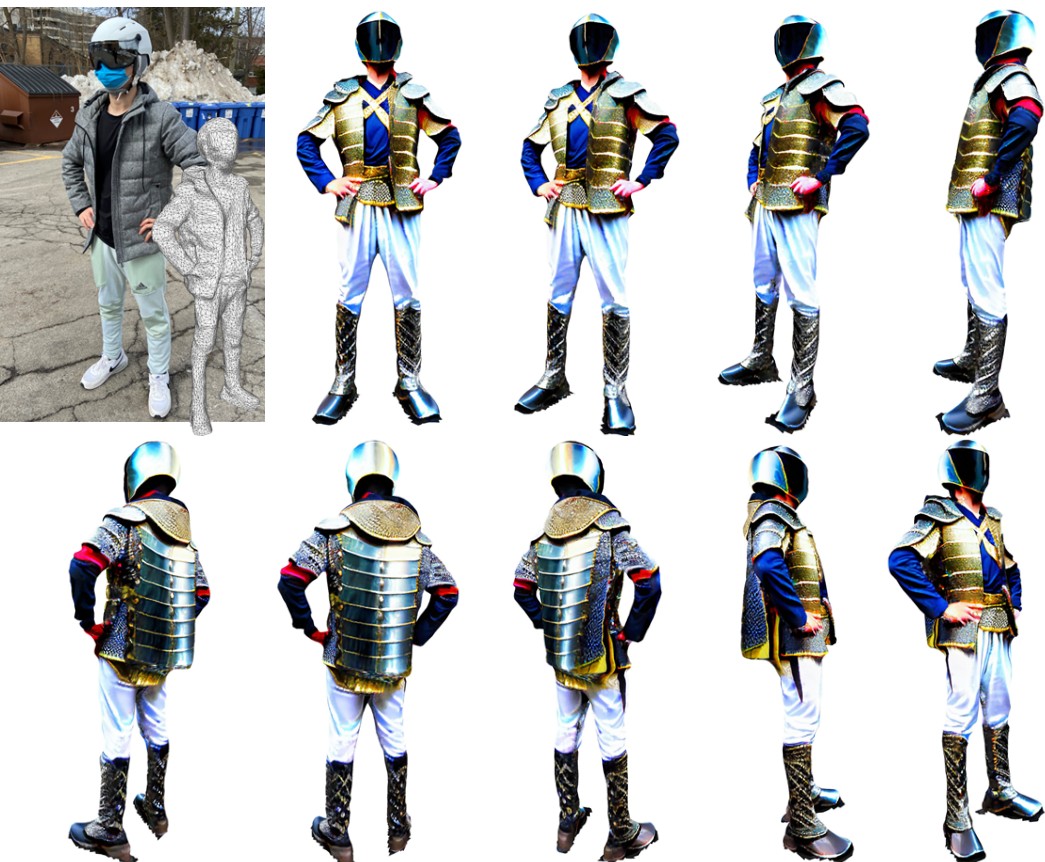

Figure 4: Results of the prompt "A man wears a cosplay suit of an armored medieval Japanese samurai".

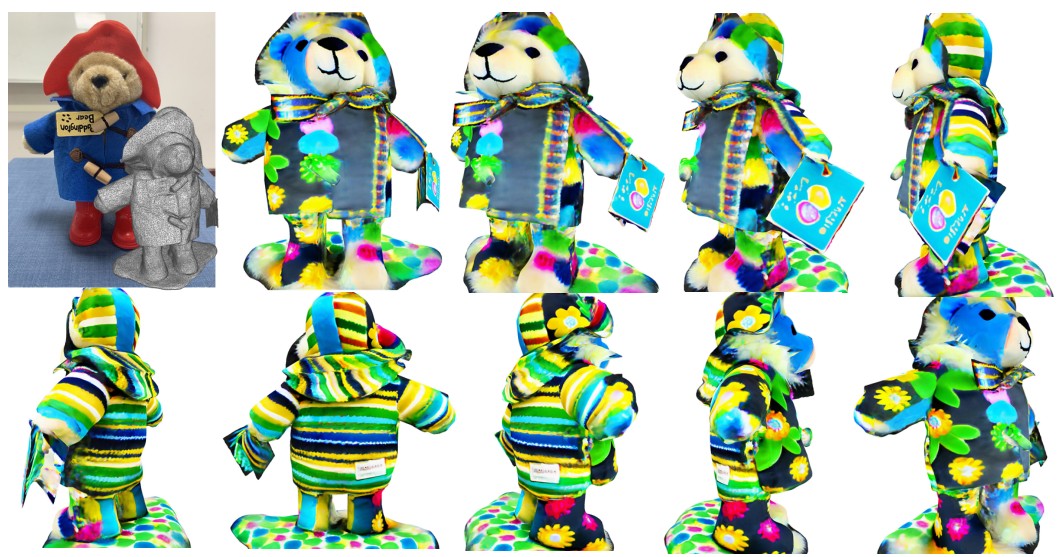

Figure 5: Results of the prompt "A plush teddy bear doll".

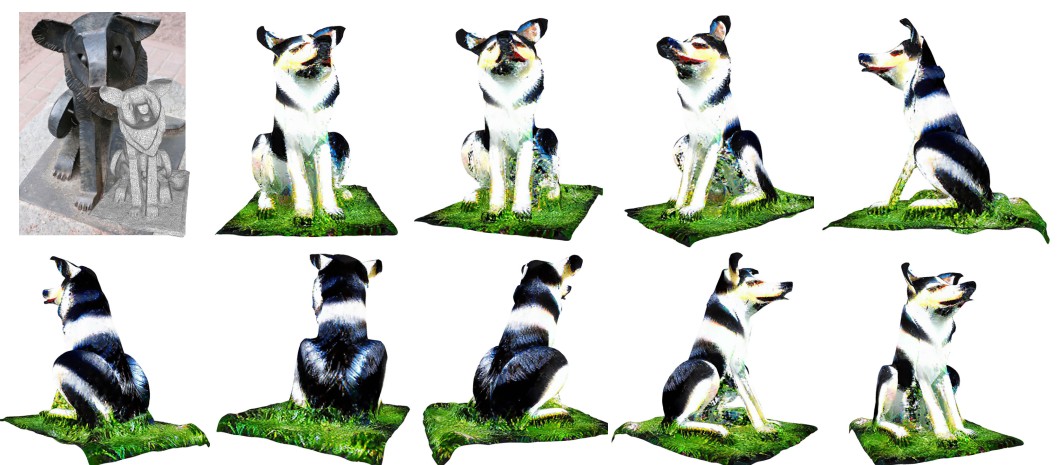

Figure 6: Results of the prompt "Husky sits on the grassland".

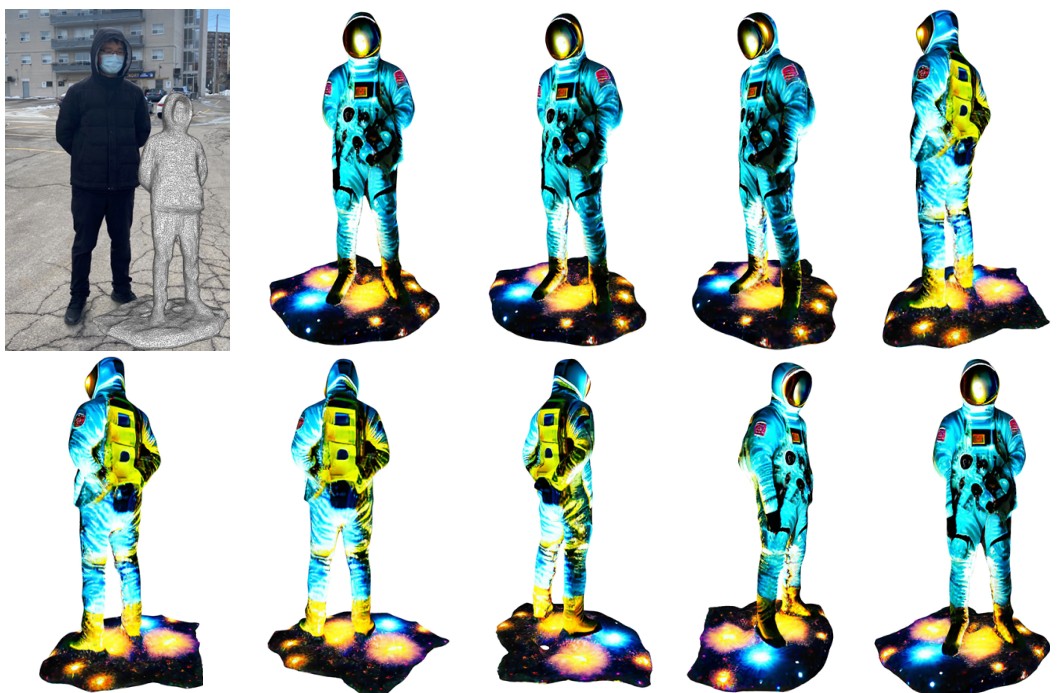

Figure 7: Results of the prompt "An astronaut stands up in the milky way".

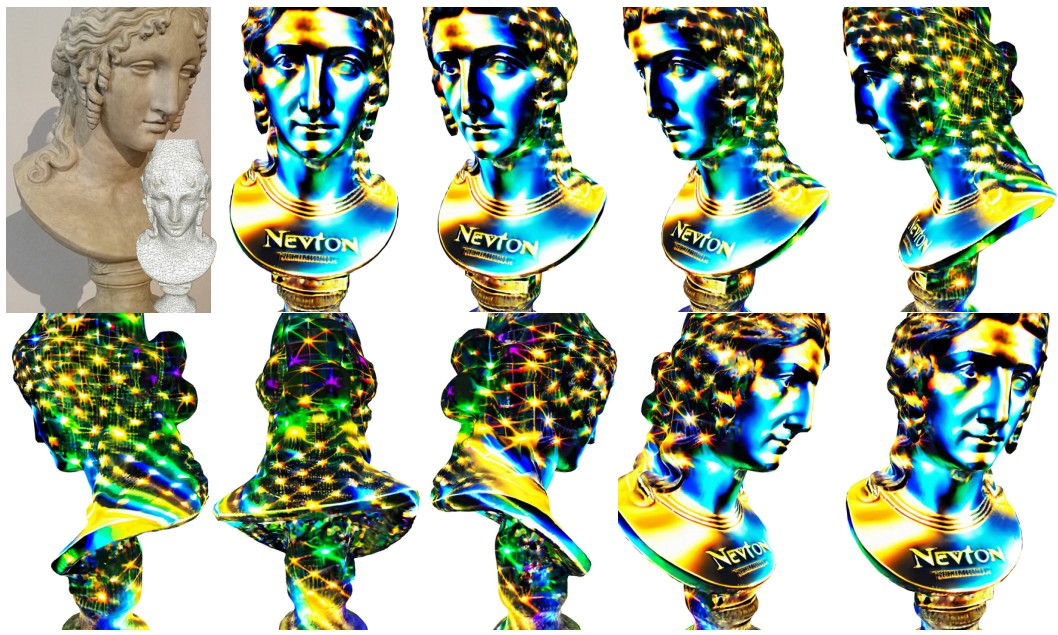

Figure 8: Results of the prompt "A close-up face sculpture of Newton".

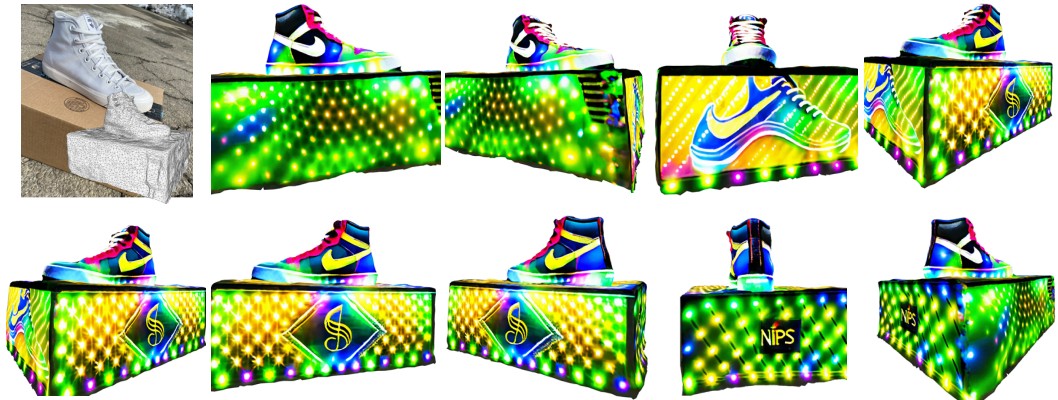

Figure 9: Results of the prompt "A sneaker pained with NIPS Logo, put on a music box".

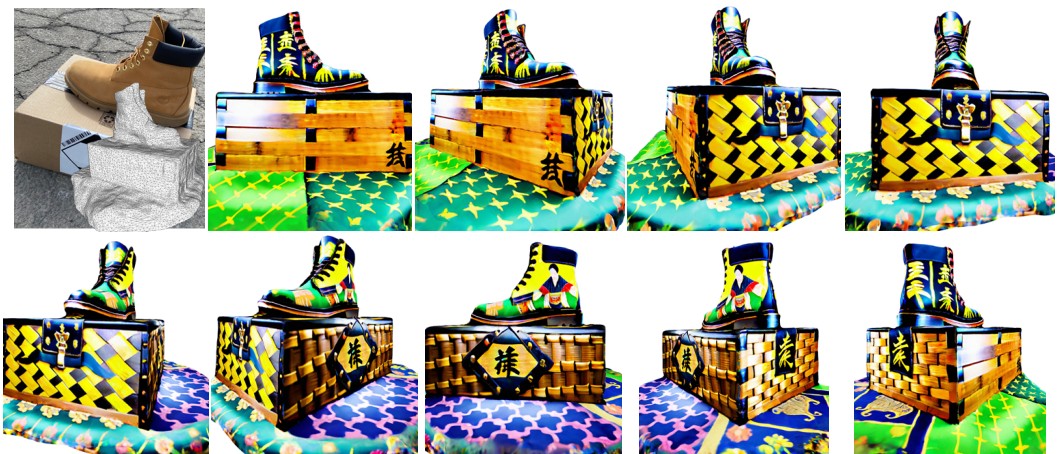

Figure 10: Results of the prompt "Man martin shoes, with samurai themes, put on a bamboo box".

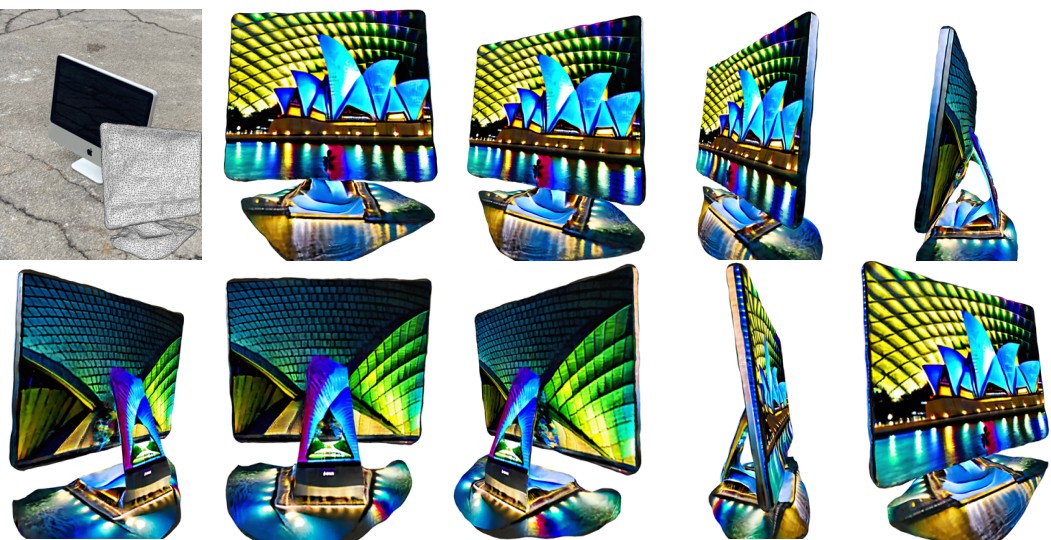

Figure 11: Results of the prompt "A monitor is displaying a photo of Sydney Opera House".

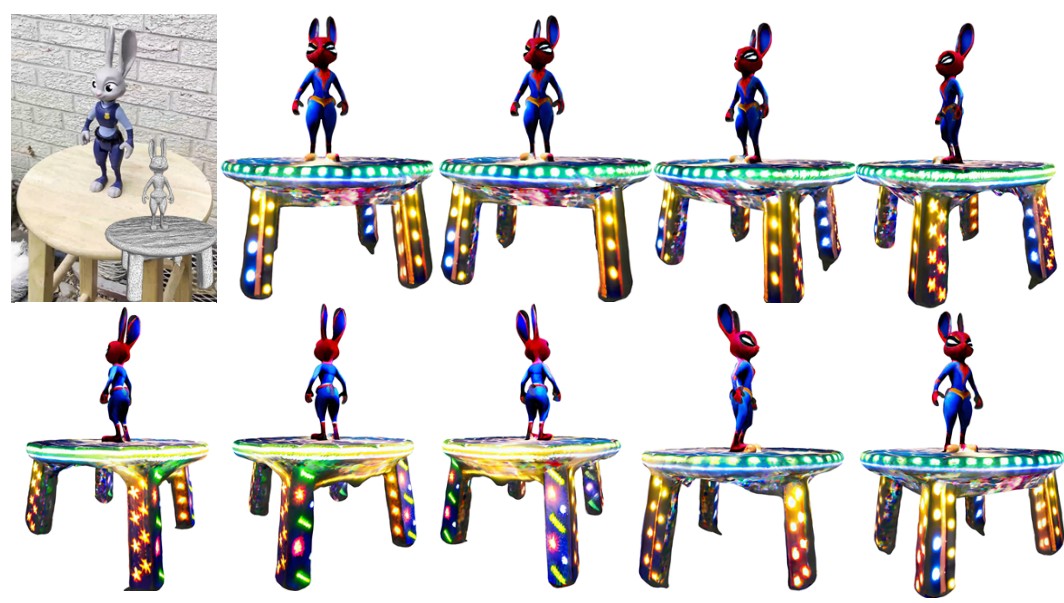

Figure 12: Results of the prompt "A character called Judy in the movie of Zootopia, wearing a Spiderman suit".

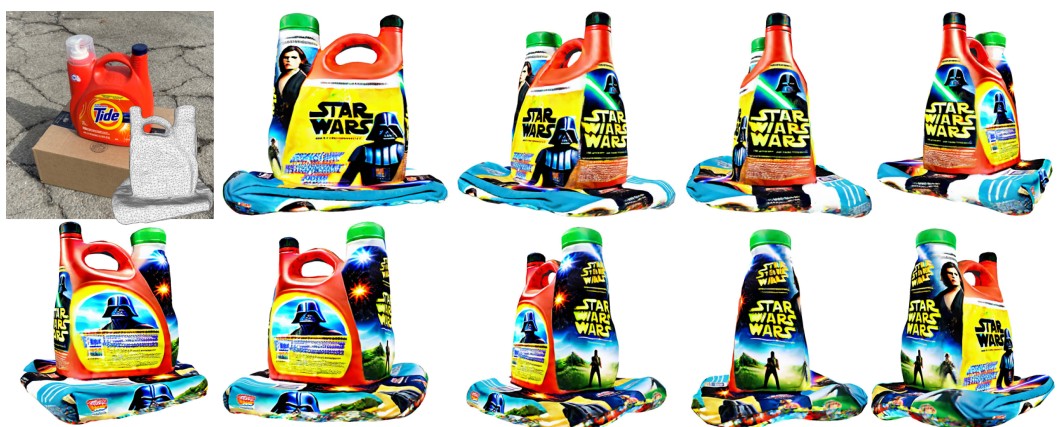

Figure 13: Results of the prompt "A detergent case, printed with a movie poster of Star Wars".

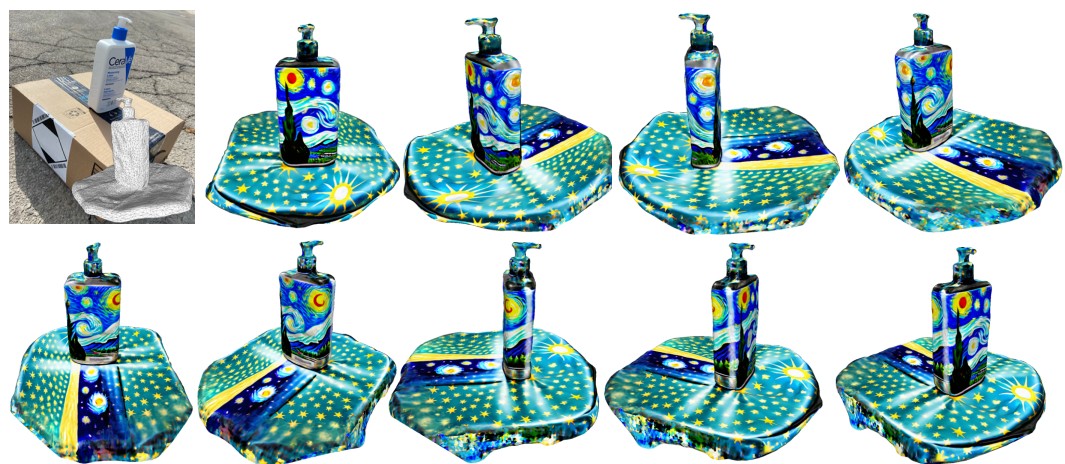

Figure 14: Results of the prompt "A lotion case, printed with Starry Night".

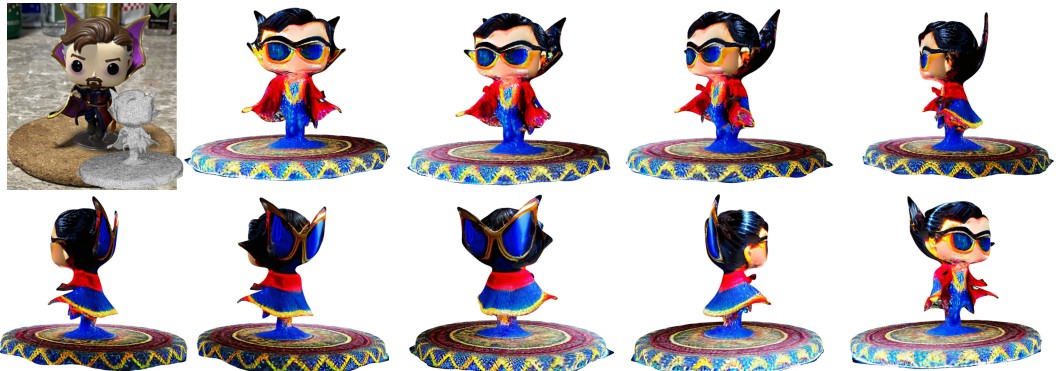

Figure 15: Results of the prompt "Dr Strange doll wears a pair of sunglasses, in a fashion style".

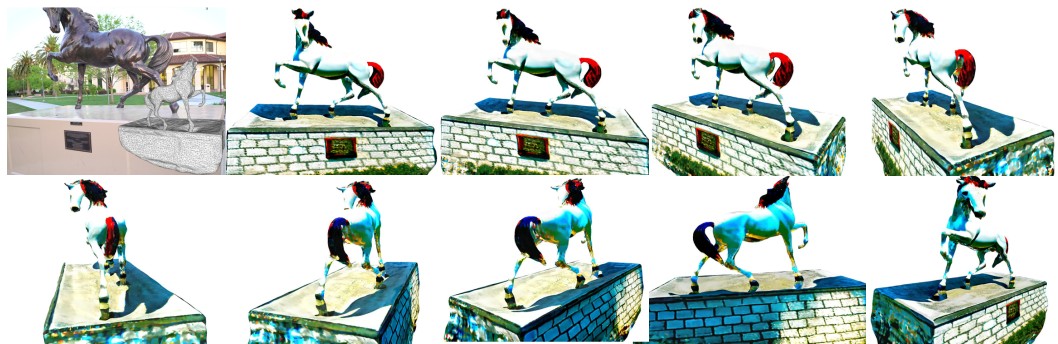

Figure 16: Results of the prompt "A white horse statue, with red hairs".

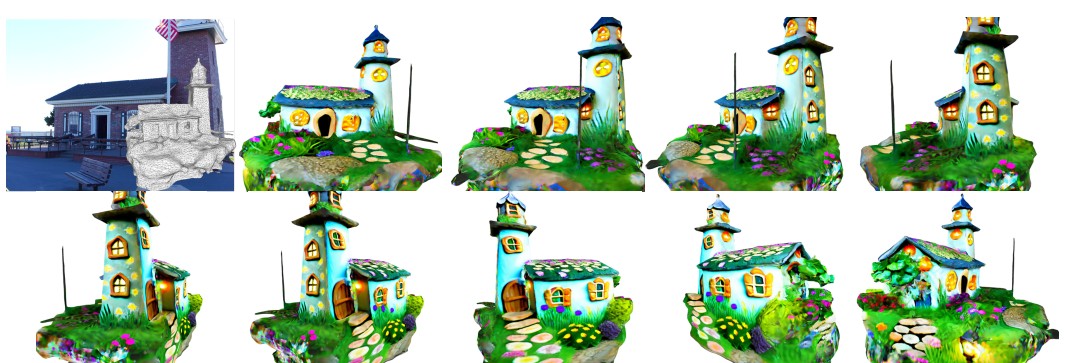

Figure 17: Results of the prompt "Fairy house with a garden".

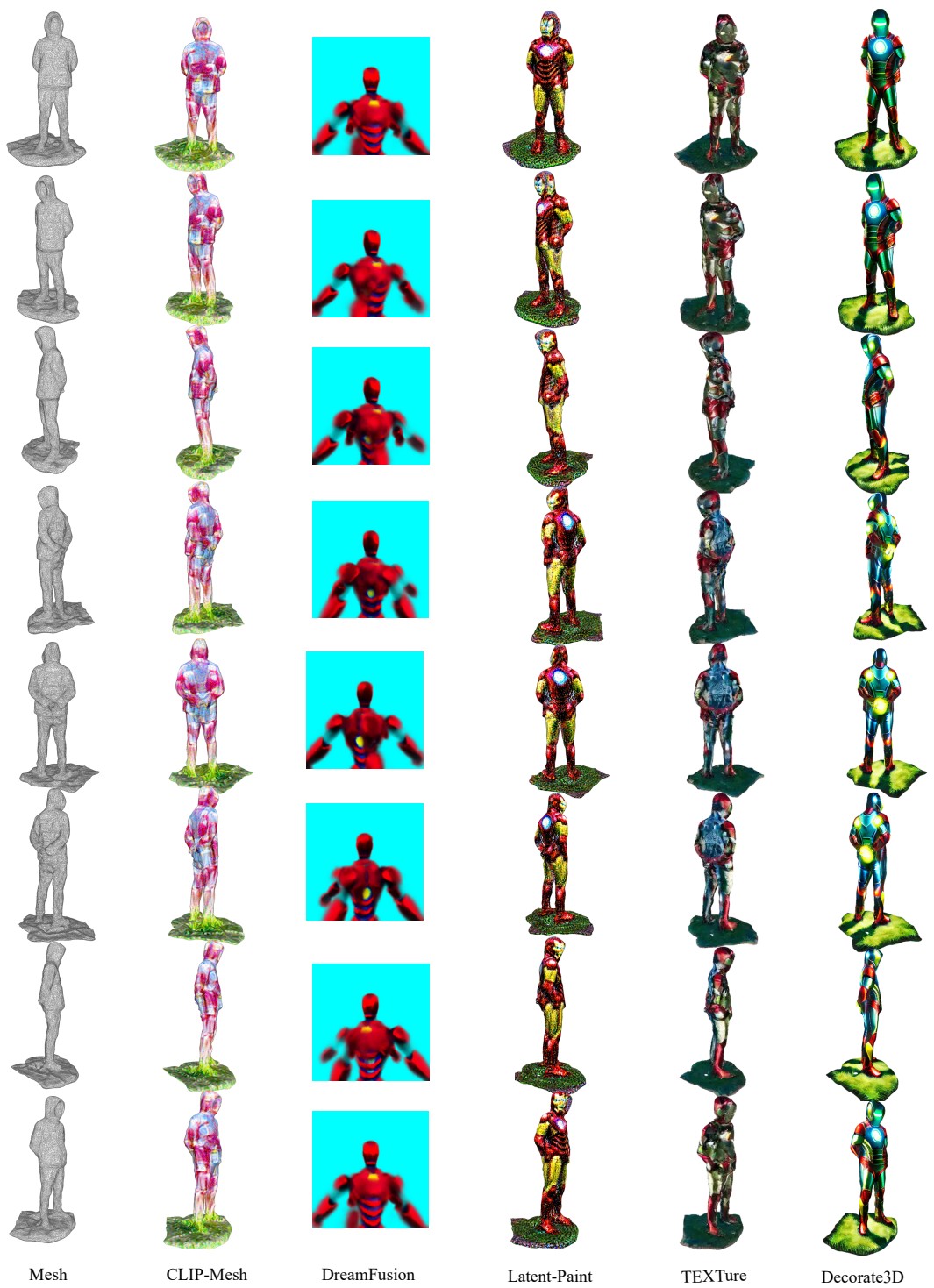

| Mesh | CLIP-Mesh | DreamFusion | Latent-Paint | TEXTure | Decorate3D |

Figure 18: A qualitative comparison between Decorate3D and the competing methods. The results are produced from the prompt "Ironman, with hands clasped behind back, standing up on the grass".

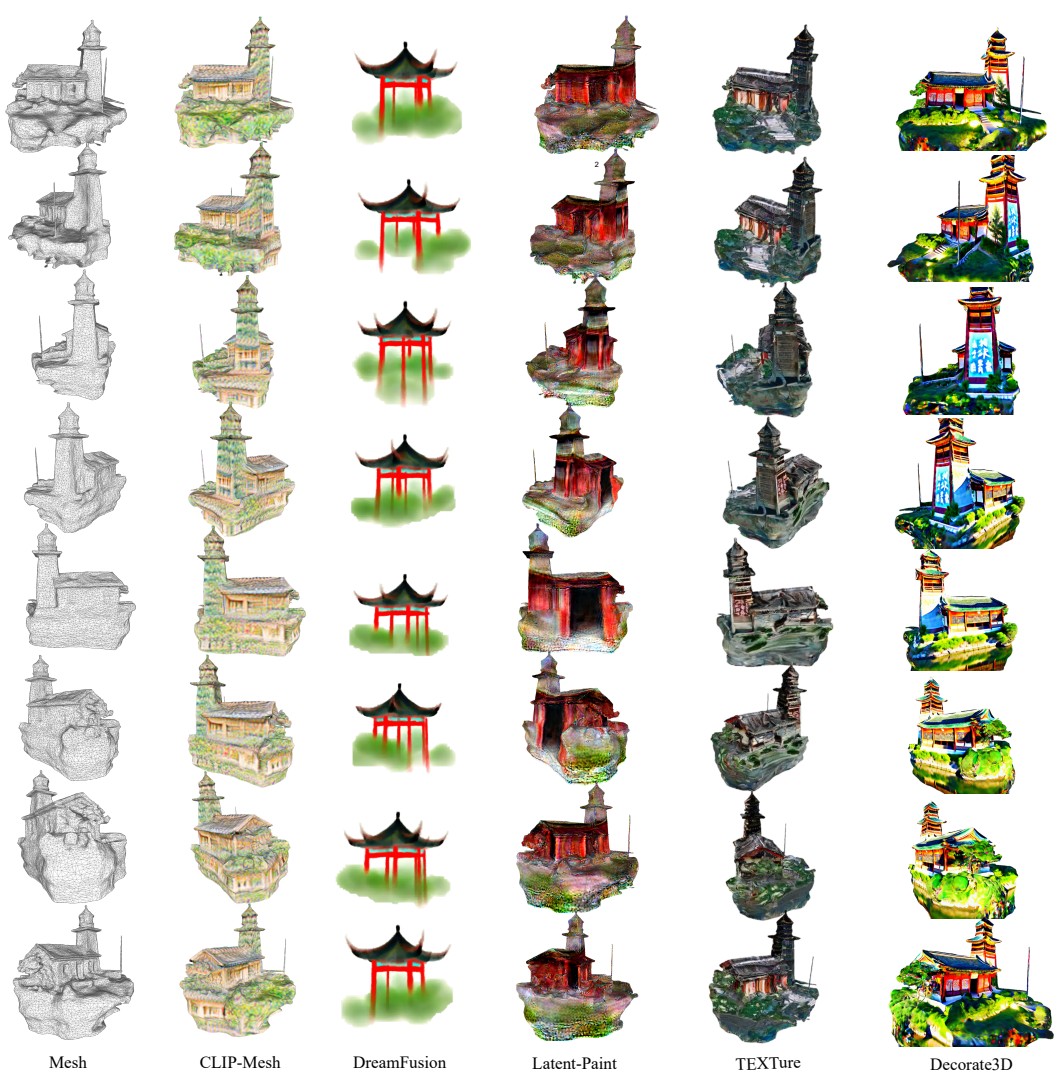

Mesh     CLIP-Mesh     DreamFusion     Latent-Paint     TEXTure     Decorate3D

Figure 19: A qualitative comparison between Decorate3D and the competing methods. The results are produced from the prompt "An ancient Chinese-style house".

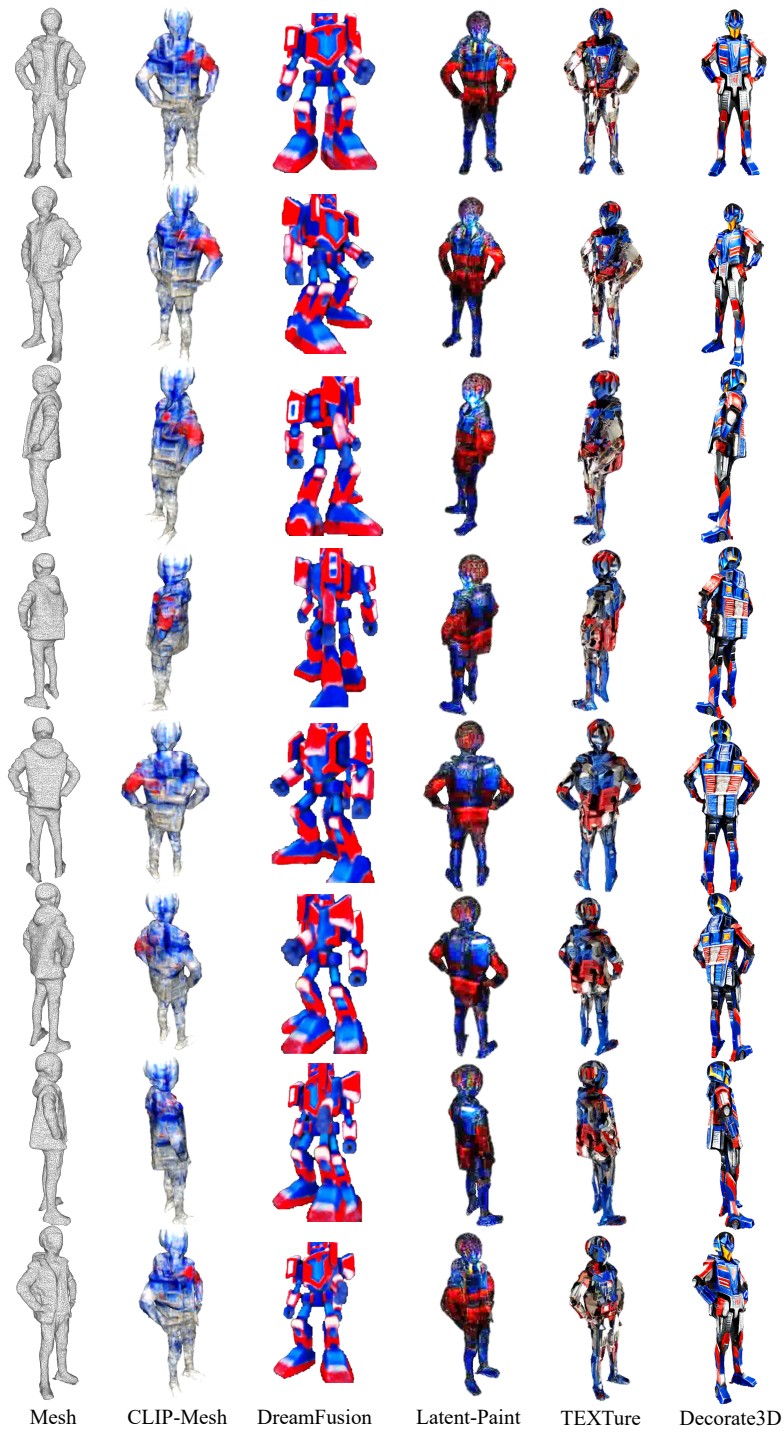

Mesh     CLIP-Mesh     DreamFusion     Latent-Paint     TEXTure     Decorate3D

Figure 20: A qualitative comparison between Decorate3D and the competing methods. The results are produced from the prompt "Optimus Prime".

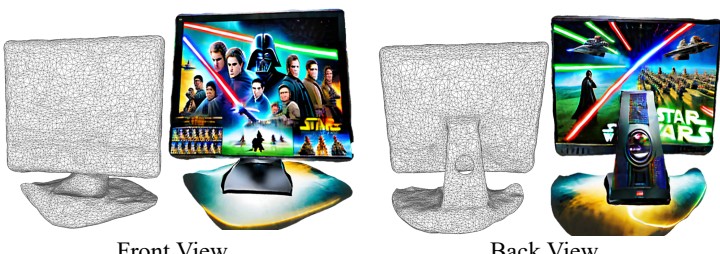

Front View                Back View

Figure 21: A failure case on the flat surface. The "monitor" object uses the prompt "A monitor is displaying the star wars film". Our method cannot correctly recognize the front and the back of the monitor's surface. As a result, both the front and the back sides are rendered with similar textures.

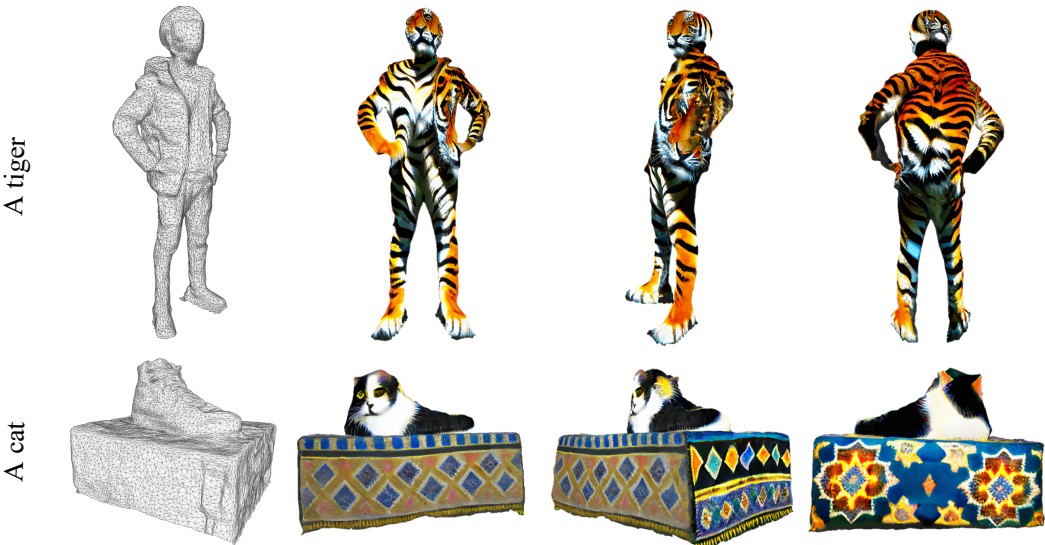

Figure 22: Two failure cases of the text-driven generation using the misaligned prompt and mesh surface. The objects "human2" and "shoes1" use the prompts "A tiger" and "A cat", respectively. Due to the mismatch between the text prompt and the mesh surface, our method cannot produce harmonious textures.