# OpenReview forum: "Decorate3D: Text-Driven High-Quality Texture Generation for Mesh Decoration in the Wild"
_NeurIPS.cc/2023/Conference — NeurIPS 2023 poster_

### Official Review · Reviewer_7jco · 2023-06-25

**Soundness:** 3 good
**Presentation:** 3 good
**Contribution:** 3 good
**Rating:** 6
**Confidence:** 3

**Summary:**

The authors present Decorate3D, a technique for text-driven texturing of a 3D mesh given a NeRF representation of a given scene. To this end, the authors introduce a two-stage texturing scheme. First, the NeRF is decomposed into a 3D mesh and view-dependent texture map. Second, given the reconstructed diffuse UV texture map, the authors edit the mesh using a modified score-distillation objective that considers the structure, or depth, of the input. Finally, to mitigate some jittering artifacts of the resulting editing texture map, the authors propose a few-view resampling technique. The authors compare their texturing scheme to existing techniques and provide many visual examples to show the 3D consistency of the textured meshes. Additional ablation studies are provided to validate the core design choices of Decorate3D.

**Strengths:**

- The visual results achieved by Decorate3D are impressive and appear to surpass existing texturing methods. Additional quantitative evaluations across numerous objects are used to further validate the effectiveness of the technique.
- I am not familiar with existing works that operate over a real 3D scene and NeRF model. For example, to the best of my understanding, most works assume that a 3D mesh is provided. Here, the authors operate in a real-world setting, which adds an additional challenge that is overcome quite nicely.
- Although the overall system is quite complex, the different components are presented quite nicely and can be understood after careful reading. The intuitions provided by the authors to motivate the different component help in understanding the design of Decorate3D.
- Finally, many ablation studies are provided to validate the different components of Decorate3D. Although I would have liked to see more visual results, I believe this can easily be added to the revision.

**Weaknesses:**

**General Points:**
- The visual results of TEXTure raise some concern on whether the method was run correctly by the authors. From my experience with the official code base, the results should be of much higher quality. In the TEXTure paper itself, the authors show an Ironman texture of a mesh and the results look far better than those presented in the paper. Moreover, in DreamAvatar [Cao et al. 2023] the authors also compare to TEXTure and achieve much better results for TEXTure. These visual results also seem to contradict the quantitative results, which placed TEXTure quite closely to Decorate3D. I want to give the authors the benefit of the doubt here, but could the authors please clarify and verify how the results for TEXTure were obtained?
- The method cannot edit the geometry, which is needed in real-world applications. Specifically, the authors do not explore the robustness of the quality of the resulting mesh. For example, does the method still work nicely if the mesh contains defects such as holes or a few faces? Moreover, the authors assume that there is some semantic relation between the prompts and the geometry in order to get reasonable results.
    - This is discussed by the authors as a limitation.

**Ablation Studies:**
- After the decomposition stage, is the resulting diffuse texture map consistent across all views? From my understanding, already at this point, the texturing should be 3D-consistent. However, I could not find results obtained after the decomposition stage (e.g., reconstruction) to verify this.
- It is difficult to assess the contribution of the structure-aware SDS from a single visual example. This also holds for the other ablations performed by the authors (e.g., the FVR training). Additional visual results, and ideally, more quantitative evaluations (e.g., as done in Table 1) would greatly assist in truly evaluating the contribution of each component.
- I had a difficult time understanding the contribution of the few-view resampling training. If I understood correctly, Figure 8 is designed to show the improvements obtained using the FVR training. Could the authors provide some additional examples that illustrate this improvement? Could the super-resolution model be applied directly to the previous result? And if we do so, would this also help with the jittering effect? That is, I am wondering whether the improved results are from the FVR or from the super-resolution model. Based on the results provided, it appears that the FVR does provide some minor improvements, but additional examples and an ablation study on applying the super-resolution model directly to the previous step would be helpful to highlight this.

**Questions:**

- Regarding the FVR stage, the authors chose to sample 8 views, if I understand correctly. Couldn’t this miss texturing areas of objects with complex areas? Does the FVR assist mainly in simpler geometries? A discussion on where to use the FVR would be beneficial since I assume that the FVR can assist differently for different geometries.
- For the 3D-consistent texture editing, wouldn’t editing the UV texture still maintain the 3D consistency? What is the intuition behind editing one of the rendered images?

**Limitations:**

The authors discuss the limitations of their method.

---

> ### Author Rebuttal · Authors · 2023-08-08
>
> Thanks for your detailed comment. We kindly remind you to check our supplymentary material that provides some video results. We think the video results are  helpful to dispel your concerns.
> ***
> * **Q1:** Concern on results of TEXTure. Could the authors please clarify and verify how the results for TEXTure were obtained?
>
>   **A1:** We use the officially released code provided by the authors of TEXTure. To make sure we have set everything up correctly, we validated the results by employing the same human mesh model shown in their paper.  Further details can be found in the accompanying one-page PDF document (see Figure C).
>
>   We also ran the demo on HuggingFace which produces similar results. The texture looks decent on the front view but the artifacts are quite obvious at the back view. This is caused by the error accumulation of their progressive texture updating strategy starting from the front view to the back. We found the same artifacts when using our experimental mesh models.
>
>   Our mesh models are even more challenging, because
>    * **(1)** it is captured from the real world, the mesh is not perfectly clean, and
>    * **(2)** the coordinates of the reconstructed object cannot be perfectly aligned with their settings of front/back view.
>
>    The supplementary material includes an anonymous project page, where we provide the human object file used in our experiments. If necessary, the reviewer could verify the TEXTure's results with our mesh file. In the anonymous page, you can click the 'code' button and the file path in the anonymous code link is 'docs/samples/qualitative/knightarmor/texture/mesh.obj'.
>
> ***
> * **Q2:** Does the method still work nicely if the mesh contains defects such as holes or a few faces?
>
>   **A2:** Since the test cases of our mesh model are reconstructed from real-world captured images, they are not guaranteed to be watertight and there are some holes around the borders (see Figure C).
>
>    Since we use the UV parameterization to represent the texture, the generated texture quality is not affected by the number of faces. For example, the mesh of the Monitor object only has a few faces(1216 faces), while the generated texture is still decent.
>
>    Please refer to Figure C in the one-page pdf document and the supplementary material.
> ***
> * **Q3:** After the decomposition stage, is the resulting diffuse texture map consistent across all views?
>
>   **A3:**  Yes, the diffuse texture is consistent across views. Please refer to the one-page pdf document and the demo video (from 3:00 to 3:20) in the supplementary material.
>
> ***
> * **Q4:** Additional visual results, and ideally, more quantitative evaluations (e.g., as done in Table 1) would greatly assist in truly evaluating the contribution of each component.
>
>   **A4:** Please refer to Table A and Figure A in the one-page  pdf document. The supplementary material provides the vieo results.
>
> ***
> * **Q5:**  Could the authors provide some additional examples that illustrate the improvement of using FVR?
>
>   **A5:**  In the one-page pdf document, we visualize the jittering problem on the left side of Figure B. The figure shows the error maps of neighboring views, where the views are aligned to a reference view by using the rendered depth. Please also refer to the supplementary video (starting from 1:50 to 2:04) to compare the results before and after FVR training.
>
> ***
> * **Q6:** Could the super-resolution model be applied directly to the previous result to address the jittering effect?
>
>   **A6:**  FVR is designed to solve the jittering problem and super-resolution is applied directly on the UV Texture to enhance the resolution of the global texture. If we neglect the FVR while directly applying super-resolution on the rendered view after the Neural Renderer, i.e. $\text{SR}(V(\mathcal{R}(\psi,\mathcal{M},\mathcal{P}_i)))$ , the jittering effects will still exist.
>
> ***
> * **Q7:** In FVR stage, choosing 8 views will miss texturing areas of objects with complex areas?
>
>   **A7:**  Yes, there could be some missing areas if not covered by the sampled views. For the objects in our experiments, we assume N=8 views (2 elevation angles chosen from \{$-20^\circ,20^\circ$\} and 4 azimuth angles uniformly sampled between [$0^\circ$, $360^\circ$]) that can cover most of the mesh surface.
>
>    Technically, for more complex geometries, we can consider adopting a more general solution. We can infer a UV mask from the UV texture to indicate which areas are overlooked by previous N views. This UV mask can be easily computed by using the camera matrices of previously sampled N views and the UV atlas. Then we can take one more step to sample more views (M views)to cover the overlooked areas of previous N views. Finally, we can train the $\text{MLP}_{\tilde{\phi}}$ with the N+M views using the FVR training.
>
> ***
> * **Q8:** For the 3D-consistent texture editing, wouldn’t editing the UV texture still maintain the 3D consistency? What is the intuition behind editing one of the rendered images?
>
>   **A8:** Yes, directly editing the UV texture will surely maintain 3D consistency. However, editing UV texture is quite inconvenient for non-professional users, particularly when the targeted region for editing becomes fragmented across non-connected parts in the UV map. Hence, compared with editing UV texture, editing rendered images is much easier to operate for general users. 3D consistency can be maintained by propagating the editing onto UV map.
>
> ***
> If the reviewer has any further concerns, we are most willing to discuss them.

---

> ### Author Response · Authors · 2023-08-16
> **Followup Response to Reviewer 7jco**
>
> Dear Reviewer 7jco:
>
> We sincerely thank you again for reviewing our paper and we appreciate your precious advice regarding additional ablations to support our proposed technique. We deeply hope that our response has properly helped in addressing your concerns, especially the comparison results of TEXTure.
>
> If there are additional questions, please do not hesitate to let us know.
>
> Best,
>
> Paper 2236 Authors

---

> > ### Comment · Reviewer_7jco · 2023-08-19
> >
> > Thanks for the clarification regarding the comparison to TEXTure. The other clarifications made by the reviewers also helped ease some of my reservations and I am therefore happy to raise my rating.

---

> > > ### Author Response · Authors · 2023-08-19
> > > **Response to Reviewer 7jco**
> > >
> > > Dear Reviewer 7jco,
> > >
> > > We are delighted to hear that your questions have been properly addressed. we'd like to thank you again for making our work stronger, and for your time and patience in reviewing our paper.
> > >
> > > Best,
> > >
> > > Paper 2236 Authors

---

### Official Review · Reviewer_U4jz · 2023-06-30

**Soundness:** 4 excellent
**Presentation:** 4 excellent
**Contribution:** 2 fair
**Rating:** 6
**Confidence:** 4

**Summary:**

This paper proposes decorate3D, a method for re-texturing real-world 3D objects using text-conditioned image diffusion models. The proposed method can be split into a 3D reconstruction phase and a re-texturing phase. In the 3D reconstruction phase, a 3D mesh is reconstructed from set of multiview images via NeuS, and a view independent texture map is distilled via differentiable rendering. In the re-texturing phase, a depth-conditioned latent diffusion model is combined with SDS to optimize the texture map. The texture map is re-rendered by passing it through the encoder-decoder of stable diffusion to remove neural artifacts, but this step introduces jittering artifacts. Jittering artifacts are then removed via optimizing a MLP through few-view resample training to reconcile view-inconsistencies. Lastly, a super-resolution diffusion model is used to up-res the produced texture map.

**Strengths:**

1. The method is technically impressive, utilizing many interesting tricks to overcome problems associated with latent diffusion models, and thereby obtaining visually impressive experimental results.
2. The presentation of the method is precise and easy to follow. Despite the many moving parts, never once did I feel a need to backtrack due to inconsistent notations or frivolous math equations.
3. The experimental procedure is detailed and well documented. One can be confident of the reproducibility of the results (as long as the authors release the real world data they've collected). The ablations are also fairely thorough, giving clear intuitions as to the effect of each component.

**Weaknesses:**

1. Novelty:
Most components utilized in this method are either well known to the literature, or straight-forward extensions of existing workflows, such as NeuS for mesh reconstruction, disentangling view-dependency via differentiable rendering of two MLPs, using depth condition for text-to-3D, and appling super-resolution diffusion models on UV textures. Though the problem of SDS with LDMs as observed in figure 3 has not been formally studied in a research paper, knowledge of this problem is folklore within the community and the proposed neural renderer solution is rather simplistic. As such, it is not clear to me whether this paper contains enough technical novelty to be impactful in the text-to-3D field.
2. Fairness of comparisons:
The experiments can be more convincing if other SDS based approaches (namely dreamfusion and latent paint) are also equipped with depth conditioned diffusion backbones instead of the vanilla backbone. I think these are sufficiently simple modification such that they can still be considered the same method, but adapted for the re-texturing task. By the same token, none of the included baselines was designed for the task of re-texturing, and the use of an initial texture provides a significant performance boost as illustrated in one of your ablations. It would be more fair if the view-independent MLP is provided as initialization for the baselines as was done for Decorate3D.

**Questions:**

1. what is the rendering model used in the SDS optimization step? Is any lighting/view-dependent effects incorporated into the rendering equation (as used in magic3d and fantasia3d) or is it purely UV based retrieval from a neural texture? Do you have any ablations on this?
2. could this method be adapted for texture synthesis by removing the initialization and using a textureless rendering of the geometry as input to the depth estimator (thereby preventing compounding artifacts between bad initial texture and bad depth estimation)?

**Limitations:**

I think the computational cost of this method is a limitation worth mentioning - it is by far the most expensive method to run versus its baselines, whose runtime ranges from seconds to tens of minutes on a single GPU, whereas Decorate3D requires hours on full 8 GPUs.
Potential negative societal impacts such as identity theft, deep fakes, and manufacturing of disinformation should be mentioned.

---------------------------------------------Post rebuttal:
I think the changes to the manuscript promised by the authors will significantly improve the delivery and message of the paper by firmly substantiating their claims regarding the effectiveness of proposed techniques with more ablations. Therefore I'm changing my suggestion to acceptance.

---

> ### Author Rebuttal · Authors · 2023-08-08
>
> * **Q1:** Novelty: Most components utilized in this method are either ..., such as NeuS for mesh reconstruction, disentangling view-dependency via differentiable rendering of two MLPs, using depth condition for text-to-3D, and applying super-resolution diffusion models on UV textures ... As such, it is not clear to me whether this paper contains enough technical novelty to be impactful in the text-to-3D field.
>
>   **A1:**  **We extend our gratitude to the reviewer for acknowledging the technical excellence of our proposed Decorate3D which achieves state-of-the-art results.**
>
>    We would like to re-emphasize here the major difference between our Decorate3D and existing retexturing techniques. Decorate3D is designed to handle noisy 3D objects derived from real-world images. In contrast, existing retexturing approaches usually work with a given ideal mesh, commonly collected from synthetic models. The real-world setting of Decorate3D gives rise to extra challenges, which cannot be addressed by a simple extension of existing workflows. For example, as shown by the ablation study, the performance of Decorate3D drops considerably without the proposed structure-aware initialization, structure-aware SDS optimization, or FVR training etc. To the best of the authors' knowledge, Decorate3D is the first method to provide a complete and effective text-driven texture generation solution, which simultaneously optimizes the high-quality and geometry-aware texture generation, for 3D objects obtained from real-world captured images.  Therefore, we have reason to believe that the work is novel and significant as appreciated at this point by other reviewers, and would like to humbly request the reviewer to reconsider the novelty issue.
>
> ***
> * **Q2:** The experiments can be more convincing if other SDS-based approaches (namely dreamfusion and latent paint) are also equipped with depth-conditioned diffusion backbones instead of the vanilla backbone. ... It would be more fair if the view-independent MLP is provided as initialization for the baselines as was done for Decorate3D.
>
>   **A2:**  First of all, we want to point out that incorporating depth-conditioned backbone can mitigate the Multi-face Janus problem, but the the visually blurry results from DreamFusion and Latent Paint may not be attributed to the lack of depth guidance. DreamFusion is optimized over NeRF and Latent Paint is optimized over the latent space. None of them works on the UV RGB texture space, which is first verified by Decorate3D. This also prevents them from using the same initialization technique as Decorate3D to solve the challenges of retexturing real-world 3D objects. We also want to mention that the latest SOTA retexturing work, i.e. TEXTure adopted the depth guidance strategy but still performed much worse than Decorate3D.
>
>   Second, our superior performance does not just come from depth-conditioned diffusion backbone and initialization. **These two proposed techniques are used to guarantee that the generated texture can match the geometry.** Our neural renderer and Few-view Resampling Training play an important role in our high-fidelity texture. **We are the first to address this problem in UV texture generation, which the competitors cannot achieve.**
>
> ***
> * **Q3:** What is the rendering model used in the SDS optimization step?
>
>   **A3:** We used the pure diffuse rendering model for UV texture without any lighting effects incorporated. This paper primarily focuses on improving the accuracy and quality of texture generation. While we agree that the utility of lighting decomposition and material modeling are useful (which will be our future work), our generated texture quality and fidelity are not affected by those factors.
> ***
> * **Q4:** Could we remove the initialization and using a textureless rendering as input to the depth estimator?
>
>   **A4:**  The depth estimator is trained over natural images, therefore using textureless rendering might not contribute to better depth estimation. If we remove the initialization, we could use the rendered depth from z-buffer which has to be normalized to feed into the depth-conditioned diffusion model. Please refer to Figure A for the results of the ablation study involving the removal of initialization.
> ***
> * **Q5:**  Computational cost and  potential negative societal impacts such as identity theft, deep fakes, and manufacturing of disinformation should be mentioned, and a discussion on societal impacts.
>
>   **A5:**  The optimization-based generation naturally has a higher computational cost than the feed-forward methods. This shortage is a common and unsolved problem of current text-driven 3D generation techniques. Reducing the computational cost remains as our future work. For the discussion on societal impacts, we will add a broader impact statement.

---

> ### Author Response · Authors · 2023-08-16
> **Followup Response to Reviewer U4jz**
>
> Dear Reviewer U4jz:
>
> We would like to thank you again for the invaluable time you dedicated to reviewing our paper. We hope that our response can address your concerns regarding  the contributions of the paper.  Please feel free to share with us if you have further questions.
>
> Best,
>
> Paper 2236 Authors

---

> ### Comment · Reviewer_U4jz · 2023-08-18
> **Response to rebuttal**
>
> My apologies for the late response.
>
> Regarding novelty, I agree that the task addressed by Decorate3D is novel in itself, and no prior works have demonstrated similarly complete pipelines for real-world capture -> reconstruction -> retexturing. However, as pointed out by another reviewer as well, the meat of the task is in the re-texturing phase, whereas the reconstruction phase is already solved reasonably well by prior works and this paper directly use existing methods (NeuS for reconstruction and Xatlas for UV parameterization) for this phase.
> Hence, while the paper indeed proposes an original pipeline with novel combination of well-known techniques, I stand by my point that the "technical novelty" of this paper, in the stricter sense, is limited. Nonetheless, I would evaluate the novelty of the paper as a net-neutral, and my bigger concern is with the question about the fairness of comparisons.
>
> Regarding fairness of comparisons, concurrent (1) and recent but prior (2,3) works have demonstrated the feasibility of obtaining sharp textures with vanilla SDS losses, without the proposed structure-aware SDS or few view resample training.
> Thus it is not clear whether structure aware SDS or few view resample training truly improve the quality of generated textures when compared to a well optimized and carefully implemented baseline adapted to this task.
> I was hoping that the authors would implement a reasonably naive baseline where depth conditioned SDS loss (with depth from render buffer) is used to optimize a surface color MLP parameterized in UV space, thereby allowing the use of the same initialization from the original texture. Instead, the authors seem to consider this as less of a baseline and more of an ablation to their method (e.g. Decorate3D minus neural renderer and FVR). If one do consider this setup as a baseline, then the relative improvement between Decorate3D and said baseline will appear smaller, and more thorough comparisons would be required than the 1-2 images currently presented in the ablation studies. I think this change would still be a net-positive on for the impact of this paper because a paper that presents a method which firmly improves upon realistic baselines will elucidate more knowledge than the paper in its current form.
>
> I also concur with reviewer xCPF's comment that this paper stands to gain more clarity if the reconstruction phase is moved out of the methods sections to focus solely on the retexturing task.
>
> An additional question I have after browsing through the project webpage is that a majority of the results have a "starry" pattern and a toy-ish palette (of sharp greens and purples), even in prompts where such textures are unexpected (such as the newton sculpture, plush teddy bear doll, and the tables under the zootopia figure). I suspected that there is interaction between initial texture and depth estimation in the original review, and Figure A's w/o init and w/o depth result seems to be in agreement with this suspicion. Do we know at which stage in the pipeline do these patterns occur and are there ways to mitigate them to produce more photorealistic textures?
>
> Lastly, an important barely concurrent (~March 2023) work (4) also addressing the problem of editing real world 3D scenes should be discussed.
>
> 1. DreamHuman: Animatable 3D Avatars from Text (https://arxiv.org/abs/2306.09329)
> 2. threestudio's implementation of Magic3D (https://github.com/threestudio-project/threestudio)
> 3. Fantasia3D: Disentangling Geometry and Appearance for High-quality Text-to-3D Content Creation (https://arxiv.org/abs/2303.13873)
> 4. Instruct-NeRF2NeRF: Editing 3D Scenes with Instructions (https://arxiv.org/abs/2303.13873)

---

> > ### Author Response · Authors · 2023-08-19
> > **Followup Response To Reviewer U4jz**
> >
> > Thanks for your continued engagement and feedback to help us improve our paper. We are delighted to hear that you have reassessed our technical novelty and acknowledged our contribution to the overall pipeline of re-texturing the real captured objects.
> > ***
> > - **Comparison with Magic3D and Fantasia3D:**
> > We uploaded extra comparisons with them using the same text prompt as ours. **The video results are available on our anonymous project homepage, where the link can be found in the supplementary pdf file.**  Since there is extra uncertainty in the geometry generation for those two methods, for fair comparison, we adjust those two approaches by initializing the DMTET [5] geometry representation with the same mesh model as in our paper and we lock the geometry as not trainable and only activate the texture generation function. It is worth mentioning that, for Magic3D, we have also incorporated the depth-conditioned SDS; and for Fantasia3D, it used the guidance model of ControlNet with Normal conditions. As you will see in the video, the generated textures from Magic3D and Fantasia3D are sharper than those produced by DreamFusion but noisier and are not as high fidelity and clear as ours.
> > Our superior performance of generating clean and clear texture owing to the proposed simple yet effective Neural Renderer and Few-View resampling Training.
> >
> > [5] Shen etal, "Deep Marching Tetrahedra: a Hybrid Representation for High-Resolution 3D Shape Synthesis", NeurIPS 2021
> >
> > ***
> > - **Contributions in Neural Renderer and Few-View Resampling (FVR) Training**
> >   - We have provided several ablations about Neural Renderer and FVR in the rebuttal. The over-saturated and noisy textures are common issues in SDS-based methods. We are the first ones who propose an effective solution. Albeit simple and straightforward, they are effective and elegant. The aforementioned comparison experiments with Magic3D and Fantasia3D as well as our extra ablations in the rebuttal, all support that our Neural Renderer and FVR are essential for generating high-fidelity textures.
> >
> >   - We agree that having additional ablations on them could be a net-positive for the impact of our paper. As suggested by reviewers, we will re-organize the structure of our paper and include more ablation study and comparison results demonstrated in the rebuttal as well as in this response. To be more specific, since it is difficult to measure the results w and w/o Neural Renderer with metrics, we will include more visual comparisons in addition to those ablations demonstrated in the rebuttal.
> >
> >   - To validate the effectiveness of FVR, we can measure the difference in pixel values between neighboring frames (which are warped onto the reference view) to any chosen reference image before and after FVR. We have included an error map (left side of Fig. B) in the one-page rebuttal document. Furthermore, in the Table below, we show the averaged quantitative results on all the cases we have tested.
> >
> >   - | Rotation | w/o FVR | w/ FVR |
> >   | :----: | :----: | ----: |
> >   | $+1^\circ$ | 0.022 | $<10^{-5}$ |
> >   | $+5^\circ$    | 0.038      | $<10^{-5}$     |
> >   | $+10^\circ$    | 0.044     | $<10^{-5}$    |
> >
> >     For each 3D model, we randomly chose a reference image, and we got its neighboring frames through the Neural Renderer by sampling a camera with a rotation angle of 1 degree, 5 degrees, and 10 degrees. For evaluation, we aligned the  neighboring frames to the reference view by a depth-guided warping and then computed the jittering errors.
> >     As shown in the above Table, without FVR, there is a pixel error between neighboring frames which will cause jittering artifacts in 360 rendered video. In contrast, after FVR, the pixel error will significantly decrease to almost 0. We neglected the occluded pixels when calculating the pixel errors.
> >
> > ***
> > - **Structure-aware SDS loss:**
> >    We believe that we all agree on the importance of having our structure-aware SDS loss which has been verified in our ablations with both quantitative evaluations and visual results. Instead of treating it as a baseline, the reasons that we think it might be worthy of mentioning this in the paper are: 1) first, to the best of our knowledge, at the time when we submitted our paper, there was not any existing work addressing this; 2) second, it will bring useful insights into the 3D generation community to realize the importance of incorporating structure constraints to the score distillation sampling.

---

> > > ### Author Response · Authors · 2023-08-19
> > > **(Continued) Followup Response To Reviewer U4jz**
> > >
> > > - **"starry'' patterns and a "toy-ish palette'' (of sharp greens and purples):**
> > >    From our experimental experience, these artifacts on the texture arise when not giving any specific text prompts corresponding to those surface regions. For example, for the cases of Zootopia and Teddy Bear, there is not any prompt describing the tables under Zootopia, or the floor of Teddy Bear standing on, This will lead to some random and periodic patterns. To elaborate more on this, you can check the results of the human models with the prompts of "Ronald McDonald'" and "Captain America stands on the desert", where the former case has the mentioned meaningless ground patterns, but the latter one has the generated ground that matches the prompt ``desert''.
> > >
> > > ***
> > > - Thanks the Reviewer U4jz for recommending several concurrent works.
> > >
> > >   1) Instruct-NeRF2NeRF [4] conducted the 3D editing in the NeRF representations. Instead of exploiting SDS losses, they progressively and explicitly replaced the multi-view images for NeRF reconstruction by adopting a pre-trained 2D image editing model called Instruct-Pix2Pix. From our understanding, the major drawback of their method is that they didn't have any pixel-wise correspondence constraints over different view directions when explicitly conducting image editing. Instead, we use a UV map to maintain pixel-wise consistency. Therefore, their rendered images from optimized NeRF are rather blurry compared with ours.
> > >   2) For Magic3D and Fantasia3D, we have provided extra comparisons on our homepage.
> > >   3) DreamHuman has just been released recently after the submission and there is not any publicly available codebase. Therefore, we cannot conduct any experimental comparisons. But from their demonstrated models in the paper as well as on their website, we do believe we still have achieved clearer texture details with much higher resolution.
> > >
> > > ***
> > > Thanks again for your suggestions. We hope our repsonse can dispel your concerns. Please do not hesitate to let us know if there are any other questions, and we are more than willing to help on them.

---

### Official Review · Reviewer_xCPF · 2023-07-03

**Soundness:** 4 excellent
**Presentation:** 2 fair
**Contribution:** 4 excellent
**Rating:** 7
**Confidence:** 3

**Summary:**

This paper proposes a method to edit the textures for neural fields (NeRFs) using score distillation sampling and also export a mesh model with texture that can be used in traditional graphics pipelines (i.e. game engines, VFX). More specifically, the main contributions that I see from this work is the "Few-view Resampling Training" which can take an SDS-optimized RGB diffuse texture map (which is noisy due to the nature of SDS with LDMs), and refine it through LDM-driven re-rendering, which takes advantage of both "LDM as a renderer" and having a real 3D consistent 3D model that can be used. This is specifically a general technique that could be widely applicable in a variety of different tasks.

In addition to this, they also create an entire pipeline to extract editable & high quality mesh representations from multi-view images (i.e. ones with good geometry, good UV parameterization, diffuse + specular separation, mostly based on existing tools) as well as another case study on SDS-driven texture generation.

**Strengths:**

The main strength of this paper is in the "few view resample training", which takes as input a noisy 3D model, renders the 3D model, refines the rendered image using a "neural renderer" (which in this case is the VAE of an LDM), and back propagates the refinement back to the 3D model. This as far as I'm aware is an original idea that I have not seen at least in this specific context. The method also seems to be effective from the limited results I am able to see, and is something that can likely be incorporated into many different contexts.

The paper also proposes an end-to-end pipeline for doing NeRF -> mesh -> editing, and evaluates several different tricks to make this pipeline effective which they also evaluate in some limited ablation studies. This is significant as it provides a case study for implementation tricks in making this pipeline work (which in my experience tends to be a big part of SDS based pipelines).

The clarity of the paper could be improved, but is not something that significantly detriments the paper. This will be discussed further in the weaknesses.

**Weaknesses:**

The biggest weakness of this paper is in its clarity. With some restructuring and refinement, however, I think that this paper could be very convincing.

First, the paper introduces the problem of 'mesh decoration'. The task really at hand is 'retexturing' or 'texture editing'. I'm not sure what the motivation for using the word 'decoration' is, but this is something that makes the paper unnecessarily confusing to grasp.

Second, the paper puts a lot of weight on discussing the end-to-end pipeline from reconstruction to retexturing. In reality, the meat of the contributions for this paper lies in the retexturing method (and specifically the few view refinement), and the rest feels like a distraction that is not core to the contribution. Making the writing and contribution statements more specific to the retexturing part of the pipeline, and treating the end-to-end pipeline as almost an 'implementation detail' would make the paper much more convincing. (i.e. the decomposition stage isn't really core to the method, since the same pipeline could be applicable for an existing 3D mesh).

Third, the paper does not sufficiently compare and contrast their method with concurrent works like TEXTure which can be considered prior art given its more than 2 months before the deadline. The paper does compare against them in the evaluations, which is great, but it could use more discussion on _why_ these prior arts produce bad artifacts in their generation, and what fundamental differences makes this paper more advantageous.

Fourth, the results shown on the core contributions are rather light. It would be very illustrative to show (on multiple models) the rendered 3D models after the SDS optimization (with their artifacts & UV textures), after "neural rendering", and after refinement to really showcase the efficacy of the refinement method.

Fifth, the ablations are good but they could be on different figures with more examples. Some of the text space that is currently used for the description of 'prior things' like SDS and the decomposition stage could probably be placed in the supplemental or taken out to make more space for results. More results on especially the effects of higher viewpoints for the refinement step could be very useful.

Lastly, it would be useful potentially to write the approximate time for completion for each stage in Figure 2 to make the costs more clear.

These are not things that affect my rating, but nitpicks:

22: "Since the implicit representations of the NeRF model are tightly coupled" this should be explained in more detail. I believe the authors are referring to the fact that it's difficult to disentangle geometry and texture from a typical NeRF model, but this phrase does not communicate this at all.

41: "The reason is that the optimized UV texture stands for neural features in effect, which produce rendered neural images that necessitate a neural interpreter" I find the whole section starting with this sentence rather confusing and hard to interpret what it really means (without having read the rest of the paper at this point yet). Trying to describe this more precisely would help. For example, what does "stands for ... in effect" mean? What is a "neural interpreter"? I can make inferences but those are then inferences. At this point I'm also confused as a reader why the UV texture needs to be 'neural' or 'latent' at all.

113: "However, diversified 3D generation is often infeasible due to a lack of enough data pairs of text and 3D models" I assume this sentence is in reference to an auto encoder sort of framework, but this is not explained anywhere. Also, I'm not really sure what "diversified" means.

131: It's not made clear in this general section that the thing that is being passed to the encoder is a rendered image, not the texture map. Although this is clear from the figure, explicitly stating this here would be nice.

The writing could be improved stylistically in various places, like not starting sentences with "And".

**Questions:**

1. Why is the task at hand called 'decoration' as opposed to texturing or texture editing?

2. At least to my eyes, it does not seem like the higher N for FVR makes a big difference. What would happen if we choose more extreme numbers for these, like N=1,2,4,8 and N=1024,2048? Is there a way to design a loss to make it robust to both extremes?

3. Is there any cases where the multi view consistency fails for FVR? How does the number of views affect this?

**Limitations:**

The authors have adequately addressed the limitations. I think they could make a broader statement on the societal impacts, as content creation tools like this are something that could potentially impact labor markets (and displace artists) and is something that is based on diffusion models trained on large amounts of data (which often also means they are unattributed / has no provenance back to artists).

---

> ### Author Rebuttal · Authors · 2023-08-08
>
> Thanks for your insightful suggestions that are helpful in improving our paper. We will carefully check and refine the descriptions that may lead to ambiguity.
>
> As suggested by the reviewer, we demonstrate more ablations of each component of the proposed pipeline and show the results in the one-page response pdf. We will also reorganize the paper structure, emphasize more on the re-texturing part, and add those ablations into the final version.
> ***
> * **Q1:** The motivation for using the word 'decoration'.
>
>   **A1:** We concur with your assessment that the essence of 'mesh decoration' lies in retexturing. However, a slight difference exists between them. Decorate3D's ultimate objective is to offer a solution for controlling or editing the texture of real-world 3D objects where a given 3D mesh is unavailable.
>
>    The proposed method successfully accomplishes the entire pipeline: 'real-world images $\rightarrow$ user editing signal $\rightarrow$ textured mesh', whereas most retexturing methods presume the availability of an ideal 3D mesh. In the real-world context, Decorate3D encounters an additional challenge and effectively overcomes it.
>
>    Additionally, from the standpoint of non-specialized users, the term 'decoration' carries greater expressiveness.  But, we will consider to revise the 'decoration phase' to 'retexturing phase' and change 'Decorate3D' to 'TextureGen3D' or 'DecoTexure3D' to improve the clarity.
>
> ***
>
> * **Q2:** Analysis and comparison with the TEXTure paper.
>
>   **A2:**  In addition to the comparison cases presented in our main paper, we have provided more results in the supplementary material file (from Figure 18 to Figure 20) and the demo video (please refer to the demo video from 0:30 to 1:30).
>
>    TEXTure proposes a progressive UV texture generation approach utilizing 10 selected views. Specifically, it initiates texture generation by first creating a front-view image through a pre-trained stable diffusion model. Subsequently, this front view is propagated onto the UV texture, and its neighboring view is generated via an inpainting model conditioned on the priorly generated texture. Consequently, TEXTure excels at generating a high-quality texture for the object's front view. However, the final generated UV texture may exhibit seams across different sampled views, and the inconsistency could accumulate when progressively updating from the front view to the back view.
>
>    In contrast to the progressive updating strategy, our UV texture undergoes global optimization through a structure-aware SDS loss. As a result, the generated texture is seamless and consistent across both the front and back views. Additionally, our FVR training significantly enhances the quality of the generated UV texture.
> ***
> * **Q3:** It would be very illustrative to show (on multiple models) the rendered 3D models after the SDS optimization (with their artifacts & UV textures), after "neural rendering", and after refinement to really showcase the efficacy of the refinement method.
>
>   **A3:** We add the results of extra quantitative and qualitative ablation studies in the one-page response pdf. In detail, Figure A demonstrates the ablations with and without initialization, depth condition as well as Neural Renderer. Figure B shows the difference map between neighboring frames before and after FVR to better visualize the jittering problem. We use ground truth depth to compute the perspective warping from neighboring frames to reference frame.
>
> ***
>
> * **Q4:** It would be useful potentially to write the approximate time for completion for each stage in Figure 2 to make the costs more clear.
>
>   **A4:** The decomposition stage takes about 3 minutes. The neural texture optimization in the decoration phase takes about 2 hours for 100K iterations, and the FVR training takes about 5 minutes.
>
> ***
> * **Q5:**  The effects of setting the number of views for FVR training. Is there a way to design a loss to make it robust to both extremes?
>
>    **A5:**   If setting a rather small number of views, like N=1,2,4, the UV texture may not be fully covered by these limited views. As illustrated on the right side of Figure B in the one-page general response PDF, we demonstrated an extreme case with N=2048. The resulting generated texture appears slightly blurry, but the difference is not easily discernible when compared with N=256 or 512. However, it should be noted that increasing the number of views will also entail additional computation costs.
>
>     In our experiments, we set N=8 empirically, which includes 2 elevation angles {$\{-20^\circ, 20^\circ\}$ }and 4 azimuth angles uniformly sampled between [$0^\circ$, $360^\circ$]. This configuration adequately covers most of the mesh surface.
>
>     As suggested by the reviewer, we can consider an algorithm that maximizes the UV coverage while at the same time minimizing the overlapping pixels in the UV.
>
>     We can deduce a UV mask from the UV texture to identify the areas that were overlooked by the previous N views. This UV mask can be easily and accurately computed using the camera matrices of previously sampled N views and the UV atlas. Subsequently, we can take an additional step by sampling more views (M views) to cover the previously overlooked areas. Ultimately, we train the $\text{MLP}_{\tilde{\phi}}$ using the FVR training with N+M views.
> ***
> * **Q6:** Is there any cases where the multi-view consistency fails for FVR? How does the number of views affect this?
>
>   **A6:** Before applying FVR, those rendered images of different views are consistent except for the jittering effects. We haven't found such cases where multi-view consistency has failed.

---

> > ### Comment · Reviewer_xCPF · 2023-08-21
> >
> > Thanks authors for providing additional details for the paper. Most of the concerns I had about the paper were having to do with clarity, which the authors commented on and gave some suggestions on ways of making it more clear. I believe including a more verbose comparison to TEXture and highlight the global optimization step would be super important for the final manuscript. I will update my score to an accept.

---

> > > ### Author Response · Authors · 2023-08-21
> > > **Thank you for the update, Reviewer xCPF!**
> > >
> > > Thank you very much for the update! We are glad that we have addressed your concerns. We'd like to thank you again for making our work stronger and for your time and patience in reviewing our paper.

---

> ### Author Response · Authors · 2023-08-16
> **Followup Response to Reviewer xCPF**
>
> Dear Reviewer xCPF:
>
> We sincerely thank you again for your great efforts in reviewing our paper, especially for the insightful suggestions on re-oragnizing the pape structure, and additional ablation analysis of each component. which effectively strengthens our paper. Please do not hesitate to let us know if there are additional questions, we would be more than happy to help with them.
>
> Best,
>
> Paper 2236 Authors

---

### Official Review · Reviewer_MiDH · 2023-07-08

**Soundness:** 3 good
**Presentation:** 4 excellent
**Contribution:** 3 good
**Rating:** 8
**Confidence:** 4

**Summary:**

This paper introduces Decorate3D, which enables text-guided 3D model editing by extracting and editing a learned UV texture. Specifically, Given multi-view images, Decorate3D first generates 3D mesh and UV textures based on NeuS. Then, it optimizes neural textures by the guidance of 3D structure (depth) and stable diffusion model. Finally,  an RGB UV texture is optimized and upsampled to generate the final result. Experiments demonstrate the state-of-the-art performance of Decorate3D.

**Strengths:**

- The idea of this paper is well-motivated and presented.

- A carefully designed pipeline (Nerf rendering, depth-aware texture optimization, few-shot texture re-optimization, and texture super-resolution) enables high-quality generation results. The 3D consistent decoration phase is novel and effective.

- Best performance is achieved compared to SOTA.

**Weaknesses:**

- The proposed method uses few-view resample training to obtain a UV texture that reduces the jittering effects. I am wondering if this step can be done in the decomposition phase or the text-driven neural texture optimization.

- The super-resolution is applied to the UV texture. However, the UV texture is not a natural image, if the model is trained with natural images, will there be some domain gap leading to inferior results? Moreover, as super-resolution is only a post-processing step that enhances the results and is not one of the main contributions, I recommend spending fewer texts on this point.

- I would like to see some quantitative ablation studies like Table 1.

**Questions:**

Some discussions about the above weaknesses and a quantitative ablation study would make the submission stronger.

**Limitations:**

Important limitations have been discussed in the paper.

---

> ### Author Rebuttal · Authors · 2023-08-08
>
>
> Thanks for your valuable comment and positive feedback. We have demonstrated extra ablations, including visual results and quantitative ablation study results. Please refer to the one-page response pdf.
> ***
> * **Q1:** The proposed method uses few-view resample training to obtain a UV texture that reduces the jittering effects. I am wondering if this step can be done in the decomposition phase or the text-driven neural texture optimization.
>
>   **A1:** The texture after decomposition is consistent, and there is no jittering problem. Therefore, it wouldn't make much difference if applying FVR in the decomposition phase. We use FVR after the text-driven neural texture optimization since the jittering problem is caused by the Neural Renderer when rendering the optimized texture across various camera views.
> ***
>
> * **Q2:** The super-resolution is applied to the UV texture. However, the UV texture is not a natural image, if the model is trained with natural images, will there be some domain gap leading to inferior results?
>
>   **A2:** The rationale behind this successful application of UV texture upscaling is that 1) the upsampling operation has a spatial locality, concentrating on local textures such as edges. For this reason, SR models are usually trained on cropped image patches instead of the whole image to increase training efficiency. This spatial locality of SR allows the SR model trained on natural images to be directly applied to the UV texture. 2) We used the SR model that is fine-tuned over the pre-trained stable diffusion, which has a strong prior to boost its generalization ability.
> ***
> As suggested by the reviewer, we will move some details of the super-resolution part into the Implementation Details.

---

> ### Author Response · Authors · 2023-08-16
> **Followup Response to Reviewer MiDH**
>
> Dear Reviewer MiDH:
>
> Thanks again for your valuable suggestions and we sincerely appreciate your acknowledgement of our work. We hope our clarifications on FVR and super-resolution can solve your concerns. Please feel free to share with us if you have any more questions.
>
> Best,
>
> Paper 2236 Authors

---

> > ### Comment · Reviewer_MiDH · 2023-08-20
> >
> > I would like to thank the authors for the rebuttal that addressed my concerns. The responses to other reviewers are also quite helpful for the audience to understand the merits of the paper. My rating remains.

---

> > > ### Author Response · Authors · 2023-08-20
> > > **Thank you, Reviewer MiDH!**
> > >
> > > We are glad to hear from you, and sincerely appreciate your acknowledgement to our work. Many thanks again for your insightful suggestions.

---

### Author Rebuttal · Authors · 2023-08-08

Thanks to all the reviewers for their constructive suggestions. Extra visual results and quantitative evaluations are included in our submitted one-page pdf document, as suggested by reviewers:
* (1) Figure A shows more ablation study results on the proposed components of Decorate3D including initialization, depth guidance, and our neural renderer;
* (2) Table A provides the quantitative ablation study results;
* (3) Figure B visualizes the error map to demonstrate the jittering artifacts and provides more ablation study results using a bigger N for FVR training;
* (4) Figure C provides extra results of TEXTure, rendering results after our decomposition phase, and the real-world meshes.

**We would like to bring to your notice that  video results have been included in our supplementary material**. It is suggested to watch them for more visually appealing results from Decorate3D.
Next, we will respond to each reviewer separately about their comments and questions.

---

### Decision · Program_Chairs · 2023-09-21

**Decision:**

Accept (poster)

**Comment:**

The paper proposed a framework to do text-guided texture generation in the wild, the idea that reviewers are excited about is mainly texture generation with a neural renderer, few-view resampling training, and a super-resolution diffusion model. These three components helped the paper to achieve better quality in texture generation (as shown in both qualitative results and the ablation studies). The rationale behind these three components can provide good insights for the community and thus, I recommend accepting the paper following the majority of the reviewers.

As promised in the rebuttal, please add more details for super-resolution modules, clarification on TEXTure comparisons, the fair comparisons with magic3d and fatasia3D, and more visualizations of the ablation studies (for neural renderer and few-view resampling training).